# Incremental BPE Tokenization

**Shenghu Jiang** [1 2]   **Ruihao Gong** [1 2]

## Abstract

We propose a novel algorithm for incremental Byte Pair Encoding (BPE) tokenization. The algorithm processes each input byte in **worst-case** $\mathcal{O}(\log^2 t)$ time, leading to an overall complexity of $\mathcal{O}(n \log^2 t)$, where $n$ is the input length and $t$ is the maximum token length. The algorithm incrementally maintains BPE tokenization results for every prefix of the input text, implementing the standard BPE merge procedure defined by a fixed set of merge rules. This enables efficient partial tokenization in streaming settings. Functioning as a drop-in replacement for standard BPE, our approach achieves a speedup of up to $\sim 3\times$ over Hugging Face's `tokenizers`, and demonstrates significant latency reductions over OpenAI's `tiktoken` on pathological inputs. We further introduce an eager output algorithm that enables streaming output, emitting tokens as soon as token boundaries are determined during incremental tokenization. Overall, our results demonstrate that BPE tokenization can be performed incrementally with strong worst-case guarantees, while providing practical latency benefits in modern large language model pipelines. The source code is available at https://github.com /ModelTC/mtc-inc-bpe.

## 1. Introduction

Byte Pair Encoding (BPE), originally proposed as a data compression technique by Gage (1994), has become the de facto tokenization method for modern large language models (LLMs) following its adaptation by Sennrich et al. (2016). By iteratively merging frequent adjacent symbol pairs, BPE constructs a compact vocabulary that balances expressiveness with computational efficiency. Consequently, highly optimized implementations such as Hugging

Email: Shenghu Jiang <research@chielo.org>. [1]Beihang University [2]SenseTime Research. Correspondence to: Ruihao Gong <gongruihao@buaa.edu.cn>.

*Proceedings of the 43rd International Conference on Machine Learning*, Seoul, South Korea. PMLR 306, 2026. Copyright 2026 by the author(s).

Face's `tokenizers` (Hugging Face, 2026) and OpenAI's `tiktoken` (OpenAI, 2026b) have become standard components in modern LLM pipelines.

Figure 1 illustrates a representative pipeline used in modern LLM tokenization, where BPE is applied as a core stage within a broader sequence of preprocessing steps.

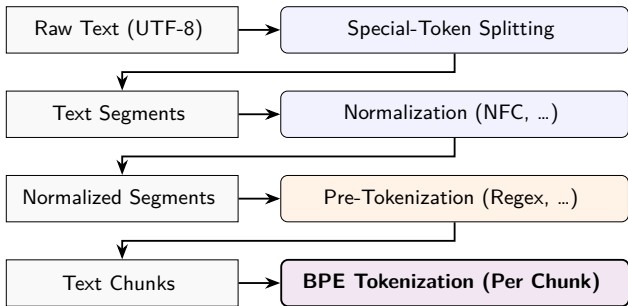

*Figure 1.* **A representative pipeline in modern LLM tokenization.** Raw text is processed by a sequence of model-specific preprocessing stages before applying BPE tokenization to each resulting chunk. *Our work focuses on making the BPE stage incremental without altering the correctness and the surrounding stages.*

However, these tools are predominantly designed as offline processes: they require the entire input sequence (or a pre-segmented chunk) to be fully observed before producing a canonical tokenization. As a result, tokenization and prefilling are executed sequentially rather than concurrently, which introduces additional latency in streaming and long-context inference.

At an algorithmic level, existing implementations typically rely on heap-based priority queues to select merges over the full input, leading to a formulation that is inherently global and difficult to adapt to byte-by-byte updates. The formalization of BPE by Berglund & van der Merwe (2023) implies a structural property where the tokenizations for every prefix of a text form a prefix tree of tokens. This property suggests the possibility of immediate feedback and partial processing; however, exploiting it efficiently requires algorithmic support beyond standard implementations.

In this paper, we present a novel algorithm for **incremental BPE tokenization** that efficiently maintains BPE tokenizations for every prefix in real-time. Our algorithm processes each input byte in a worst-case time complex-

ity of $\mathcal{O}(\log^2 t)$, resulting in an overall time complexity of $\mathcal{O}(n \log^2 t)$, where $n$ is the input length and $t$ is the maximum token length.

We implement the algorithm in Rust as a drop-in replacement for core BPE stages. Benchmark results (Table 1) show an end-to-end speedup of up to $\sim 3\times$ over Hugging Face's `tokenizers`. Furthermore, while existing tools like OpenAI's `tiktoken` exhibit $\mathcal{O}(n^2)$ behavior on certain inputs, our approach maintains a stable $\mathcal{O}(n \log^2 t)$ cost, leading to significant latency reductions on pathological cases (Figure 3).

Beyond raw speed and streaming input, we introduce an *eager output* mechanism for streaming output, which emits tokens as soon as stable token boundaries are resolved. This enables tokenization to be fully pipelined with model inference.

Profiling of current systems (Appendix I) shows that stages other than BPE can become bottlenecks in tokenization pipelines, burdening both streaming input and output. With strict worst-case complexity guarantees, such pre-segmentation can be algorithmically redundant.

Furthermore, along with incremental updates, the algorithm naturally supports scenarios requiring full-prefix access, i.e., scenarios in which tokenizations of all prefixes are required, such as deriving attention masks from the prefix tree of tokens for "fill-in-the-middle" (FIM) tasks without random or heuristic truncation and re-computation.

Our results demonstrate that incremental BPE tokenization is both algorithmically feasible and practically beneficial, with **strict worst-case guarantees** and **exact semantic compatibility**.

**Contributions.** We make the following contributions:

- An incremental BPE algorithm is proposed with a strict worst-case $\mathcal{O}(\log^2 t)$ per-byte complexity, ensuring exact equivalence with standard BPE alongside rigorous performance guarantees.

- The proposed eager output mechanism enables efficient, real-time streaming and pipelining with model inference.

- Our efficient implementation in Rust serves as a drop-in replacement that delivers a speedup of up to $\sim 3\times$ over current state-of-the-art tools.

- With our method, stages of traditional pipelines that serve solely to circumvent the limits of global BPE can be safely removed without performance concerns.

- Our method provides native support for full-prefix tasks (e.g., FIM, token healing) by efficiently providing tokenizations of all prefixes, eliminating the need for heuristic truncation or re-computation.

## 2. Related Work

Byte Pair Encoding (Sennrich et al., 2016) has established itself as the de facto standard for modern LLMs, employed by state-of-the-art architectures such as GPT-5 (OpenAI, 2026a) and Qwen-3 (Qwen Team, 2025a). Despite alternatives like WordPiece (Schuster & Nakajima, 2012) used in BERT (Devlin et al., 2019) and Unigram (Kudo, 2018) adopted by T5 (Raffel et al., 2020), BPE remains the predominant choice for current models.

In terms of model architecture, usage patterns have evolved: while earlier models typically applied BPE to the unsegmented input text, modern LLMs increasingly adopt regex-based pre-segmentation and other normalization to enforce linguistic boundaries and partition the input.

Leading libraries implement these strategies differently: Hugging Face's `tokenizers` (Hugging Face, 2026) provides a general-purpose implementation using priority queues, whereas OpenAI's `tiktoken` (OpenAI, 2026b) specializes in the regex-based approach, relying on segmentation to bound the cost of BPE merges.

Despite these optimizations, production-grade tokenization remains predominantly *offline*. Whether relying on global queues or pre-segmentation, canonical BPE tokenization is typically performed after a complete segment is available.

This architecture restricts the ability to hide latency by pipelining tokenization with model inference at a fine-grained level. Furthermore, the reliance on regex engines for complexity control can be brittle, potentially introducing performance degradation or stack exhaustion on pathological inputs such as extensive repetitions.

On the theoretical side, Berglund & van der Merwe (2023) analyze the formal semantics of different BPE implementations and note its consistency under token-wise truncation. This observation implies that the tokenization of a string prefix remains stable within the full tokenization, hinting at the possibility of incremental processing. However, their work focuses on algebraic properties rather than algorithmic construction, and does not address the challenge of bounding the lookahead required for online updates.

In the engineering domain, practical implementations such as the crate `bpe` in `rust-gems` (GitHub, 2026), whose techniques are discussed by van Antwerpen & Neubeck (2024), have adopted the Aho–Corasick automaton (Aho & Corasick, 1975) for incremental tokenization. The detailed comparison is discussed in Appendix J. While demonstrating practical utility, these approaches generally lack formal worst-case complexity guarantees.

Our work bridges this gap by providing an incremental algorithm with $\mathcal{O}(\log^2 t)$ worst-case update time while maintaining strict compatibility with standard BPE.

# 3. Structural Foundations of Incremental BPE

In this section, we establish the structural foundations of our algorithm. We first formalize the incremental tokenization problem by explicitly defining the concept of the **Last Token** and deriving the recursive properties of BPE. We then introduce a normalized representation of the BPE vocabulary to eliminate redundancy. Finally, we construct the **Successor Forest** and the **Suffix-Successor Tree**, which organize the search space for incremental tokenization. These structures provide the topological basis for the theoretical properties and search algorithms presented in subsequent sections.

## 3.1. Preliminaries and Definitions

Let $\Sigma$ be a finite alphabet and $V \subset \Sigma^+$ be a finite vocabulary. The vocabulary $V$ consists of **atomic tokens** (where $|t| = 1$) and **non-atomic tokens** (where $|t| > 1$).

A **suffix token** of a string $s$ is defined as any token $t \in V$ that is a suffix of $s$.

A **token sequence** $\varphi = [t_1, \ldots, t_n]$ maps back to the string via **detokenization** $\pi(\varphi) = t_1 \ldots t_n$.

A **dictionary** is defined by an ordered list of merge rules $D = [r_1, r_2, \ldots, r_m]$. Each rule $r_i$ specifies a pair of adjacent tokens $(x, y)$ to be merged into a new token $z = xy$.

Applying a rule $r$ to a token sequence $\varphi$, denoted as $\mathbb{T}_r(\varphi)$, replaces occurrences of adjacent tokens $x$ and $y$ with $z$ sequentially from left to right.

The full BPE tokenization for a string $s$ with the dictionary $D$, denoted as $\mathbb{T}_D(s)$, is obtained by initially mapping $s$ to a sequence of characters $\varphi_0$, and then applying the rules in $D$ according to their priority:

$$\mathbb{T}_D(s) = (\mathbb{T}_{r_m} \circ \cdots \circ \mathbb{T}_{r_2} \circ \mathbb{T}_{r_1})(\varphi_0) \qquad (1)$$

where $r_1$ represents the highest priority rule.

Note that under this definition, the standard BPE differs from the SentencePiece (Kudo & Richardson, 2018) semantics. Our analysis is formulated under the standard BPE merge semantics. See Appendix A for a detailed discussion on the difference between the two tokenization semantics and the method to "properize" merge rules for the standard BPE.

In the remainder of this paper, we omit the subscript $D$ and write $\mathbb{T}(s)$ when the dictionary is clear from context.

## 3.2. Problem Formulation

**Prefix Consistency.** Our incremental approach relies on the consistency of BPE tokenization under token-wise truncation. As noted by Berglund & van der Merwe (2023)

(Remark 3), a valid BPE tokenization $\mathbb{T}(\cdot)$ can be freely truncated: the remaining sequence remains the valid tokenization of its corresponding substring.

**Lemma 3.1** (Prefix Consistency)**.** *Let $s$ be a string and $\varphi = \mathbb{T}(s)$ be its token sequence after tokenization. For any proper prefix $\mu \subset \varphi$ of the token sequence, the following identity holds:* $\mathbb{T}(\pi(\mu)) = \mu$.

Intuitively, this lemma holds because each BPE merge step is a **boundary elimination** process visualized in Figure 4. We provide an independent formal proof in Appendix B.

This lemma ensures that the tokenization of a growing string can be modeled as extending the previous valid tokenization of some shorter prefix rather than re-computing from scratch.

**The Last Token.** Based on this consistency, we define the **Last Token**, denoted as $\theta(s)$. For any non-empty string $s$, $\theta(s)$ is the last token in its BPE tokenization sequence: $\theta(s) = \text{last}(\mathbb{T}(s))$.

According to the Prefix Consistency lemma (Lemma 3.1), since removing the last token leaves a valid tokenization sequence for the remaining string prefix, we can express the tokenization recursively:

$$\mathbb{T}(s) = \mathbb{T}(s_{\text{pre}}) \oplus [\theta(s)] \qquad (2)$$

where $s_{\text{pre}}$ is the string $s$ with the suffix string of $\theta(s)$ removed, and $\oplus$ denotes sequence concatenation.

In terms of complexity, the full-prefix tokenization requires maintaining $\Theta(|s|)$ states.

**Prefix Tree of Tokens.** This recursive relationship implies that the tokenizations of all prefixes naturally form a hierarchical structure. Strictly speaking, this structure constitutes a forest, as a tokenization sequence may begin with any valid token. To unify this into a single **Prefix Tree of Tokens**, we introduce a **virtual root** representing the tokenization of the empty string $\varepsilon$, which serves as the shared root for all initial tokens.

While we visualize the space of all prefix tokenizations as this tree, strictly maintaining the explicit structure is unnecessary. Instead, the tokenization for any prefix can be retrieved solely by *backtracking* using $\theta(\cdot)$ until the beginning of the string (the virtual root) is reached.

**Incremental Objective.** Consider an incremental scenario where a character $c$ is appended to the current string $s$, resulting in $s' = sc$. To update the tokenization, we only need to find the value of $\theta(s')$. The objective is to identify $\theta(s')$ within the *intersection* of the set of all suffixes of $s'$ and the vocabulary $V$. In the following sections, we introduce the

**Successor Forest** and other structures to efficiently solve this search problem.

### 3.3. Normalization and Token Hierarchy

In practice, BPE vocabularies often contain unreachable entries. To facilitate efficient updates, we define a **normalized representation**. We say a token $t \in V$ is **canonical** if $\mathbb{T}_D(t) = [t]$. The **normalized vocabulary** $\mathcal{V}$ consists solely of canonical tokens. Crucially, the determinism of BPE establishes a **bijective mapping** between non-atomic canonical tokens and the specific merge rules that produce them, which we term **canonical rules**. The **normalized dictionary** $\mathcal{D}$ contains strictly these rules. We show in Appendix C that this normalization preserves exact compatibility, i.e., $\mathbb{T}_{\mathcal{D}}(s) \equiv \mathbb{T}_D(s)$.

**Predecessors and Successors.** Based on the bijection above, for every non-atomic $t \in \mathcal{V}$ produced by the canonical rule $(x, y)$ of $t$, we define its **predecessor** $\mathrm{pre}(t) = x$ and **successor** $\mathrm{suc}(t) = y$. As detailed in Appendix C, the determinism of BPE ensures $x, y \in \mathcal{V}$. Thus, the vocabulary is closed under these operations, forming the topological basis for the Successor Forest.

### 3.4. Successor Forest

Based on the hierarchy, we construct the topological structure for searching $\theta(\cdot)$. We define the **Successor Forest** ($\mathcal{F}_{\mathrm{suc}}$) as a directed graph $G = (\mathcal{V}, E)$ where edges point from a token to its successor: $E = \{(u, \mathrm{suc}(u)) \mid u \in \mathcal{V}, |u| > 1\}$. Since edges strictly reduce token length ($|\mathrm{suc}(u)| < |u|$), $G$ is acyclic and forms a forest of trees rooted at atomic tokens.

An example Successor Forest is illustrated in Figure 2.

### 3.5. Suffix-Successor Tree

By definition, for any non-atomic token $u$, $u = \mathrm{pre}(u)\,\mathrm{suc}(u)$, which implies $\mathrm{suc}(u)$ is a proper suffix of $u$. This property ensures that traversing parents in $\mathcal{F}_{\mathrm{suc}}$ corresponds to taking proper suffixes.

For a given token $t \in \mathcal{V}$, we define the **Suffix-Successor Tree**, denoted as $\mathrm{SufSucTree}(t)$, as the subgraph of $\mathcal{F}_{\mathrm{suc}}$ induced by the set of all canonical tokens that are suffixes of $t$.

Let $S_t$ be the set of suffix tokens of $t$ (note that $t \in S_t$). If a non-atomic token $u \in S_t$, then its parent in the forest, $\mathrm{suc}(u)$, must also be a suffix of $t$, implying $\mathrm{suc}(u) \in S_t$. Since all suffixes of $t$ share the same last character, all paths in this subgraph converge to the unique atomic suffix token of $t$. Therefore, $\mathrm{SufSucTree}(t)$ forms a single tree rooted at this atomic token.

This structure organizes all potential candidates for the value of $\theta(\cdot)$. By identifying further properties on this tree, we can reformulate the incremental tokenization as an efficient tree search problem.

An example Suffix-Successor Tree is highlighted in violet in Figure 2.

## 4. The Monotonic Path Property

This section establishes the key property underlying BPE for suffix tokens. For any input string, we will show that the suffix tokens capable of serving as the final token form a single monotonic path in the Suffix-Successor Tree.

### 4.1. Intuition

Fix a non-empty string $s$ and a suffix token $t$ of $s$. For $t$ to be the valid last token $\theta(s)$, the boundaries around its components $\mathrm{pre}(t)$ and $\mathrm{suc}(t)$ are required to survive the merge process before applying the canonical rule of $t$ (as formalized in Lemma E.3 and Lemma E.6).

This survival establishes a necessary dual constraint on the merge history:

1. **Structural Reachability**: $\mathrm{pre}(t)$ must appear as the rightmost token at some stage during the processing of the prefix, meaning the eventual last token of the prefix must be a descendant of $\mathrm{pre}(t)$.

2. **Procedural Priority**: The merge rule forming $t$ must execute *before* any rule that would otherwise consume $\mathrm{pre}(t)$ (specifically, merging it with a left neighbor).

Crucially, these conditions are necessary but not sufficient, since the newly formed token $t$ may still be involved in subsequent merges before the final tokenization is reached.

Another intuition is that the rightmost token can be viewed as evolving through the tokenization: starting from an atomic token, it is progressively merged with its adjacent token. This process can be viewed as traversing a descendant path in the Successor Forest, until reaching the last token of the final tokenization.

We provide a detailed derivation of this intuition based on the "boundary elimination" logic in Appendix D.

### 4.2. Formalization

We now formalize the intuition from the previous subsection. The following definition establishes a rigorous criterion for determining whether a candidate suffix token $t$ is compatible with the tokenization of the remaining prefix.

**Definition 4.1** (Prefix Last-Token Condition)**.** Fix a non-empty string $s$ and a suffix token $t$ of $s$.

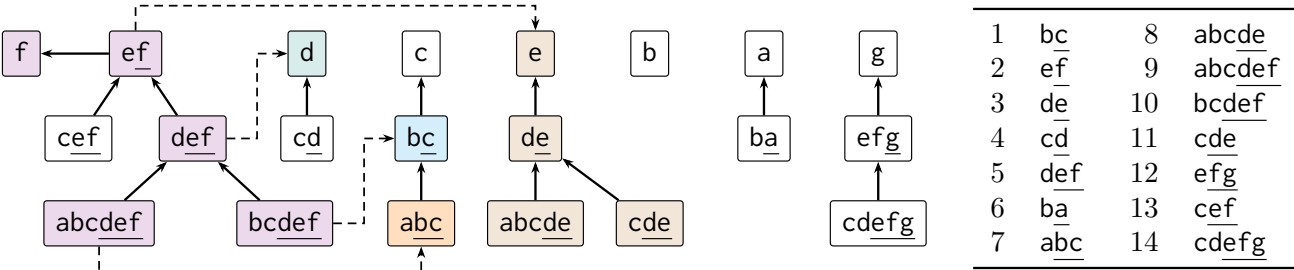

*Figure 2.* **Successor Forest and the merge rules.** Solid edges denote the Successor Forest, while dashed edges indicate predecessors. Violet nodes highlight SufSucTree(abcdef). Other colored nodes, formed by the predecessors and some of their subtrees, identify the tokens $\theta(s^{-\mathrm{suc}(t)})$ satisfying the **Prefix Last-Token Condition** (Definition 4.1) for the tokens $t$: ef, def, bcdef, and abcdef.

If $t$ is an **atomic token**, it satisfies the **Prefix Last-Token Condition** by definition.

If $t$ is a **non-atomic token**, let $s^{-\mathrm{suc}(t)}$ denote the prefix obtained by removing the suffix $\mathrm{suc}(t)$ from $s$. We say that $t$ satisfies the condition with respect to $s$ if the following hold:

1. **Reachability.** The token $\theta(s^{-\mathrm{suc}(t)})$ lies in the subtree of the Successor Forest rooted at $\mathrm{pre}(t)$ (including the case $\theta(s^{-\mathrm{suc}(t)}) = \mathrm{pre}(t)$).

2. **Priority dominance.** If $\theta(s^{-\mathrm{suc}(t)}) \neq \mathrm{pre}(t)$, let $u$ be the unique child of $\mathrm{pre}(t)$ that lies on the ancestor path from $\theta(s^{-\mathrm{suc}(t)})$ to $\mathrm{pre}(t)$. Then the canonical rule of $t$ has strictly higher priority than that of $u$.

We now assert the main structural theorem of this paper.

**Theorem 4.2** (Monotonic Path Property). *Let $k$ be the longest canonical suffix token of a string $s$. Consider the Suffix-Successor Tree* SufSucTree($k$). *Let $P$ denote the unique path from the last token $\theta(s)$ to the root. Then a node $t \in$ SufSucTree($k$) satisfies the Prefix Last-Token Condition if and only if $t$ lies on the unique path $P$.*

We provide the complete proof in Appendix E. The "if" direction follows directly from the intuition above: along the successor chain of the true last token, both reachability and priority dominance are preserved monotonically.

It is notable that, although the Monotonic Path Property restricts the search space to the Suffix-Successor Tree of the longest suffix token, *the Prefix Last-Token Condition itself is evaluated in the global Successor Forest*. In particular, for a candidate token $t$, the condition depends only on the relative position between $\mathrm{pre}(t)$ and $\theta(s^{-\mathrm{suc}(t)})$ in the Successor Forest, and does not depend on the membership of $\mathrm{pre}(t)$ in SufSucTree($t$).

### 4.3. Linearization via DFS

To utilize the Monotonic Path Property to find the last token $\theta(s)$, it requires an efficient method to verify whether a node $t \in$ SufSucTree($k$) satisfies the Prefix Last-Token Condition (Definition 4.1).

For a fixed non-atomic token $t$, consider the possible values of $\theta(s^{-\mathrm{suc}(t)})$ that satisfy the condition. This set of valid tokens forms the subtree rooted at $\mathrm{pre}(t)$ in the Successor Forest dictated by the "Reachability" requirement, excluding the subtrees rooted at any direct child of $\mathrm{pre}(t)$ whose canonical rule has a priority greater than or equal to that of $t$, as enforced by the "Priority dominance".

Let $\mathcal{C}_t$ be the set described above for a non-atomic token $t$. If $t$ is an atomic token, $\mathcal{C}_t$ is simply defined as an empty set. In this section, we show that membership in $\mathcal{C}_t$ can be evaluated in $\mathcal{O}(1)$ time with DFS linearization.

**DFS Linearization of the Successor Forest.** To linearize the forest topology, we perform a pre-order Depth-First Search (DFS) starting from the atomic tokens. We maintain a global counter that increments exactly once upon entering each node $u$, assigning it a unique entry timestamp $\mathrm{dfs\_in}(u)$. We then assign an exit timestamp $\mathrm{dfs\_out}(u)$ immediately after fully exploring its subtree.

This process establishes a bijection between the nodes and the timestamps. Crucially, it guarantees that the subtree rooted at any node $u$ corresponds exactly to a contiguous half-open interval $[\mathrm{dfs\_in}(u), \mathrm{dfs\_out}(u))$.

Additionally, when traversing the children of a node $u$, we visit them in strictly increasing order of their canonical rule priorities (**from lowest to highest priority**). This deliberate ordering ensures that children with lower priorities are mapped to earlier timestamps, enabling us to efficiently exclude the subtrees of higher-priority children to enforce the "Priority dominance" condition.

**Valid Interval.** Notably, the set $\mathcal{C}_t$ corresponds to a contiguous range of timestamps. For each non-atomic canonical token $t$, we precompute a *valid interval* $I_t = [L_t, R_t)$ such that a node $x \in \mathcal{C}_t$ if and only if $\mathrm{dfs\_in}(x) \in I_t$. Concretely:

- $L_t = \mathrm{dfs\_in}(\mathrm{pre}(t))$;

- $R_t = \text{dfs\_in}(u)$, where $u$ is the first child of $\text{pre}(t)$ (in the DFS traversal order) whose canonical rule priority is greater than or equal to that of $t$. If no such child exists, we set $R_t = \text{dfs\_out}(\text{pre}(t))$.

The construction above reduces the evaluation of the Prefix Last-Token Condition to an $\mathcal{O}(1)$ interval membership test.

**Mutual Exclusion among Siblings.** Finally, it is notable that the valid intervals of sibling nodes in the Suffix-Successor Tree are necessarily disjoint. This is a direct corollary of the Monotonic Path Property (Theorem 4.2). If the intervals of two distinct siblings overlapped, there would exist a prefix state for which both siblings satisfy the Prefix Last-Token Condition, implying the existence of valid nodes on two diverging branches. This contradicts the uniqueness of the valid path guaranteed by Theorem 4.2.

Thus, for any two distinct children $u$ and $v$ of a node, we have $I_u \cap I_v = \emptyset$. This mutual exclusion enables the search algorithm to unambiguously identify the correct branch at each step.

# 5. Incremental Algorithm

Based on the theoretical foundations established in the previous sections, we now present the incremental algorithm for maintaining the last token $\theta(s)$. In this section, we first outline the abstract algorithmic framework and its design objectives. We then detail the specific data structures, the Aho–Corasick automaton (Aho & Corasick, 1975) and Centroid Decomposition, employed to implement each component efficiently.

## 5.1. Algorithm Framework

The incremental update problem can be framed as follows: given the current string $s$, its last token $\theta(s)$, and a new character $c$, we aim to compute the new last token $\theta(sc)$.

Our strategy relies on the structural properties of the search space. First, recall that $\theta(sc)$ must be a **suffix token** of the string $sc$. While there may be many suffix tokens, they are all suffixes of the **longest suffix token**, denoted as $\tau(sc)$. Consequently, the Suffix-Successor Tree $\text{SufSucTree}(\tau(sc))$ defines our **search space**.

Within this search space, the **Monotonic Path Property** (Theorem 4.2) guarantees that the set of nodes satisfying the Prefix Last-Token Condition forms a single continuous path anchored at the root (the atomic token $c$). Since the atomic token is always a valid candidate, this path is never empty. The correct last token $\theta(sc)$ corresponds to the **deepest node** of this valid path.

Therefore, the algorithm proceeds in two abstract steps:

1. **Search Space Identification**: Identify the longest suffix token $\tau(sc)$ to determine the bounding tree $\text{SufSucTree}(\tau(sc))$.

2. **Path Extension Search**: Within $\text{SufSucTree}(\tau(sc))$, search for the deepest node satisfying the Prefix Last-Token Condition.

To evaluate the Prefix Last-Token Condition during the search, the algorithm requires access to the last tokens of previous prefixes. We abstract this as a **history interface** $\mathcal{H}(\ell)$, which returns the last token of the prefix obtained by removing $\ell$ characters from the end of the current string.

## 5.2. Search Space Maintenance via Aho–Corasick Automaton

To identify $\tau(sc)$, we maintain an Aho–Corasick automaton (Aho & Corasick, 1975) on $\mathcal{V}$. We augment the structure by precomputing the search space entry point: for each state $u$, the annotation is the token $u$ itself if $u \in \mathcal{V}$, otherwise it is inherited from $u$'s suffix link. This allows retrieving $\tau(sc)$ in $\mathcal{O}(1)$ without traversing suffix links. We store the transition table using a persistent square-root tiling (details in Appendix F) to ensure $\mathcal{O}(1)$ transition queries with memory efficiency. The annotation at the new state identifies the specific $\text{SufSucTree}(\cdot)$ for the subsequent search.

## 5.3. Efficient Search via Centroid Decomposition

The main challenge is to efficiently locate the deepest valid node within $\text{SufSucTree}(\tau)$, where $\tau = \tau(sc)$. Since the tree height can be linear in $|\tau|$, a naive top-down traversal is inefficient. We solve this by guiding the search using **Centroid Decomposition**.

For each canonical token $\tau \in \mathcal{V}$, we precompute a **Centroid Search Tree** (CST). The CST is a tree of height $\mathcal{O}(\log |\tau|)$, where each node corresponds to a centroid $u$ of a weak component in the recursive decomposition of $\text{SufSucTree}(\tau)$. Crucially, from the perspective of a centroid $u$, the original tree is split into disjoint weakly connected components: one component containing the parent of $u$ (the "upward" direction), and several components each containing a child of $u$ (the "downward" directions).

The search for $\theta(sc)$ proceeds by traversing this CST. At each step, let $u$ be the current centroid. We evaluate the Prefix Last-Token Condition for $u$ by querying the history state $k = \mathcal{H}(|\text{suc}(u)|)$ and performing the interval check $\text{dfs\_in}(k) \in I_u$. Based on the Monotonic Path Property, we determine the direction of the valid path's endpoint:

1. **Case 1: $u$ is invalid ($\text{dfs\_in}(k) \notin I_u$).** Since the valid path is anchored at the tree root (the atomic token), an invalid node $u$ implies that the path does not in-

tersect with the subtree of $\mathrm{SufSucTree}(\tau)$ rooted at $u$; i.e., the path must be inside the component of the parent. We transition to the CST child corresponding to the component containing $u$'s parent (the upward direction).

2. **Case 2:** $u$ **is valid** ($\mathrm{dfs\_in}(k) \in I_u$). The node $u$ lies on the valid path. The target $\theta(sc)$ is either $u$ itself or its descendant. To determine if the path extends deeper, we check the children of $u$ in the original $\mathrm{SufSucTree}(\tau)$. Recall from Section 4.3 that the valid intervals of siblings are disjoint. We can thus pre-sort the intervals and perform a binary search over the children to check if any child $v$ is valid.

   - If a valid child $v$ exists (i.e., $\mathrm{dfs\_in}(\mathcal{H}(|u|)) \in I_v$), the path continues into $v$'s subtree. We transition to the CST child corresponding to the component containing $v$.
   - If no child is valid, then the path ends at $u$. We conclude $\theta(sc) = u$ and terminate the search.

### 5.4. Complexity Analysis

The worst-case time complexity per byte is dominated by the centroid search.

- **Automaton State Update**: The optimized automaton transition takes $\mathcal{O}(1)$ time.

- **CST Traversal**: The CST height is bounded by $\mathcal{O}(\log|\tau|)$.

- **Decision Step**: At each CST node, we perform either a single interval check or a binary search over the children. Since the degree of a node in $\mathrm{SufSucTree}(\tau)$ is bounded by $|\tau|$, the binary search takes $\mathcal{O}(\log|\tau|)$.

Each check involves a history query $\mathcal{H}(\cdot)$, assumed to be an $\mathcal{O}(1)$ operation. Combining these factors, the complexity per byte is $\mathcal{O}(\log^2|\tau|)$, leading to an overall complexity of $\mathcal{O}(n\log^2 t)$ for the entire input, where $t$ is the maximum token length.

## 6. Eager Output

The incremental algorithm efficiently maintains the state $\theta(s)$. Since the sequence of last tokens implicitly encodes the *Prefix Tree of Tokens*, the full tokenization is typically recovered by **backtracking** from $\theta(s)$ to the virtual root.

In a streaming setting, however, incrementality alone does not guarantee streaming output. Thus, the challenge is to identify the **initial segment** of this backtracking chain that has become invariant, serving as the common ancestry for any future extension of the string.

### 6.1. The Active Frontier

The ambiguity of the output stream arises from the uncertainty of which current node in the Prefix Tree of Tokens will serve as the parent for future tokens. When new characters are appended to $s$, any newly formed last token must attach to the last token of some prefix of $s$.

The range of such possible attachments is bounded by the maximum suffix length queried during the incremental search, which corresponds to the deepest state reached in the Aho–Corasick automaton. Let $d(s)$ be the **depth** of the state reached by the string $s$ in the Aho–Corasick automaton. Note that this depth corresponds exactly to the length of the longest suffix of $s$ recognized by the automaton. Since any future token formed must be recognized by the automaton, its length cannot exceed $d(s)$.

Consequently, the parent of any future token must be the last token of a prefix ending in the interval $[|s| - d(s), |s|]$.

We thus define the set of **Parental Candidates**, $\mathcal{P}$, as the set of all last tokens that fall within this interval:

$$\mathcal{P} = \{\theta(s[1\ldots i]) \mid |s| - d(s) \le i \le |s|\} \qquad (3)$$

where $s[1\ldots 0]$ indicates the empty string $\varepsilon$ and $\theta(\varepsilon)$ is the virtual root of the Prefix Tree of Tokens. The stable output is the **common ancestral path** shared by all tokens in $\mathcal{P}$.

### 6.2. Maintenance Algorithm

To efficiently identify and emit the stable tokens, we exploit the monotonicity of the window boundaries. Observe that the automaton transition for a new character can increase the state depth by at most 1 (i.e., $d(sc) \le d(s) + 1$). Therefore, the start of the window, $|s| - d(s)$, is monotonically non-decreasing. This guarantees that candidates only "expire" from $\mathcal{P}$ as the window slides forward; previously expired tokens never re-enter.

Based on this property, we implement a two-pointer algorithm to maintain the set $\mathcal{P}$ and track the active subgraph in the Prefix Tree of Tokens. Tokens are emitted eagerly once all active paths converge to a single child of the virtual root. The detailed algorithmic steps and complexity analysis are provided in Appendix G.

While eager output enables streaming, it introduces computational overhead due to maintaining the dynamic subgraph. Therefore, unless otherwise specified, the standard benchmarks presented later utilize the non-eager incremental algorithm.

## 7. Benchmarks

We evaluate the end-to-end throughput of our incremental BPE algorithm by integrating it as a drop-in replace-

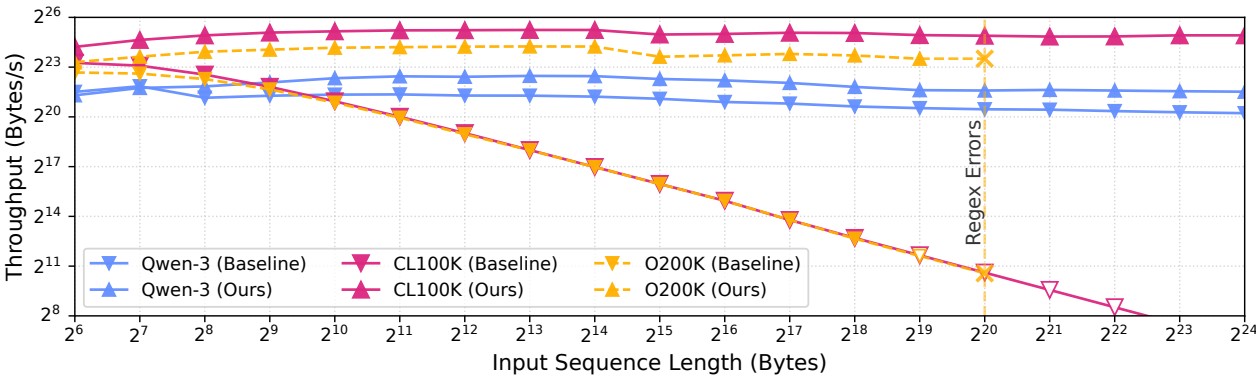

*Figure 3.* **End-to-end throughput under pathological inputs (log–log scale).** Filled markers represent measured results, while hollow markers indicate extrapolated values beyond the measurable range. The dashed **vertical** line highlights the input length at which the regular expression matching fails before BPE. Our approach demonstrates higher and more stable throughput compared to baseline methods. Notably, `tiktoken` exhibits a characteristic decay as input length increases, reflecting its underlying $\mathcal{O}(n^2)$ complexity.

ment for Hugging Face's `tokenizers` and OpenAI's `tiktoken`. Since our algorithm is fully compatible with existing pipelines, the benchmarks isolate the effect of replacing the BPE stage alone, without altering segmentation, normalization, or caching behavior.

Detailed experimental setup, dataset composition, and execution strategies are provided in Appendix H, while the **full benchmark results** (including eager output) are listed in Table 5.

### 7.1. Throughput Analysis

Table 1 reports the speedup factors evaluated on concatenated documents from each dataset.

**Absence of Pre-tokenization.**   The most substantial gains are observed in **CodeLlama** (up to **3.13×**), where the BPE processes normalized text without regex-based pre-tokenization. In this regime, the heap-based implementation in `tokenizers` faces log-linear complexity over the full input length, whereas our algorithm leverages its incremental nature to maintain high throughput.

**Coarse-Grained Segmentation.**   For the **Chinese** dataset, regex-based pre-tokenization typically yields coarser segments than English due to the specific designs of the regex rules and the characteristics of the language.

This exposes the performance bottlenecks in the original BPE implementation of `tiktoken`, while our method achieves significant speedups (up to 1.59×) by strictly bounding the per-byte processing time.

**Fine-Grained Segmentation.**   For the **English** dataset, regex rules effectively partition the input into small chunks, where the constant factors of the implementation overhead

become dominant. This explains the slight regression of our method observed in some cases.

*Table 1.* **End-to-end throughput speedup.** We report the speedup factor of our incremental BPE as a drop-in replacement for Hugging Face's `tokenizers` and OpenAI's `tiktoken` (using default configurations with short-string caches enabled where applicable). CodeLlama shows the highest gains due to the absence of regex-based pre-tokenization.

| | Tokenizer Model | English | Chinese | Code |
|---|---|---|---|---|
| `tokenizers` | CodeLlama | **3.13×** | **1.10×** | **2.88×** |
| | Qwen-3 | **1.05×** | **1.04×** | **1.08×** |
| | DeepSeek-3.2 | **1.01×** | 0.93× | **1.03×** |
| | Ouro | 1.00× | **1.02×** | **1.02×** |
| | Llama-3.1[*] | 0.99× | **1.03×** | **1.02×** |
| | GPT-OSS | 1.00× | **1.08×** | **1.01×** |
| | Llama-4 | **1.01×** | **1.01×** | 1.00× |
| | Mistral-3 | 1.00× | **1.04×** | 0.99× |
| `tiktoken` | P50K | 0.97× | **1.35×** | **1.07×** |
| | R50K | 0.96× | **1.35×** | **1.05×** |
| | CL100K | 0.96× | **1.59×** | **1.04×** |
| | O200K | 0.99× | **1.46×** | 1.00× |

[*] Properized dictionary. See Appendix A for discussion on compatibility and non-properizable cases (e.g., Gemma-3).

### 7.2. Robustness and Eager Output

**Pathological Inputs.**   We test robustness using inputs constructed by repeating the character 'a' $2^k$ times. As shown in Figure 3 and Figure 7, `tiktoken` exhibits a characteristic throughput decay consistent with $\mathcal{O}(n^2)$ complexity.

Our method maintains stable throughput throughout the tests. The other tested tokenizers within the same frameworks exhibit similar trend characteristics to the represen-

tative models shown. Notably, specifically for the O200K model as an exception, long inputs trigger errors in the regex stage before BPE.

**Eager Output Overhead.** We also evaluate our *eager output* implementation (Section 6). Results in Appendix H indicate that eager output introduces an overhead on the order of 10% in end-to-end throughput compared to the non-eager incremental implementation. This overhead stems from the additional bookkeeping required to maintain output states, and can potentially be amortized in systems where tokenization is pipelined with I/O or model inference.

## 8. Future Work

While this work demonstrates that the BPE stage itself is fully compatible with streaming input and output, other components in existing pipelines, notably normalization and regex-based pre-tokenization, remain offline-oriented and impede end-to-end streaming execution.

Our profiling results in Appendix I indicate that these upstream stages can become computational bottlenecks.

A critical direction for future work is therefore to **revisit the design choices of pre-tokenization** in light of incremental algorithms with explicit worst-case guarantees.

## 9. Conclusion

We introduced a novel algorithm for incremental BPE tokenization with a strict **worst-case time complexity** of $\mathcal{O}(n \log^2 t)$. Implemented as a drop-in replacement, the algorithm delivers substantial end-to-end speedups over existing systems and avoids the severe performance degradation on pathological inputs. Beyond performance, our results show that BPE can be made compatible with streaming input and output, enabling optimizations in modern language model systems.

Overall, our approach proves that strict algorithmic worst-case guarantees can translate into practical benefits, paving the way for more robust and responsive LLM pipelines.

## Acknowledgements

This work was supported by the National Natural Science Foundation of China (Nos. 62525601, 62476018) and the Postdoctoral Fellowship Program of CPSF (No. BX20250487).

## Impact Statement

This paper presents work whose goal is to advance the field of machine learning by improving the fundamental algorithmic infrastructure of large language models (LLMs).

**Energy Efficiency.** Tokenization is a ubiquitous preprocessing step executed trillions of times in modern LLM pipelines. By reducing the algorithmic worst-case time complexity of BPE to $\mathcal{O}(n \log^2 t)$, utilizing incremental updates for streaming input, and enabling eager emission for streaming output, our approach reduces the computational resources and latency required for processing. This optimization contributes to the broader goal of "Green AI," potentially lowering the aggregate energy consumption and carbon footprint associated with deploying large-scale language models.

**System Robustness and Reliability.** Our work specifically addresses performance degradation on pathological inputs, a vulnerability present in some current state-of-the-art tokenizers. By providing strict worst-case execution guarantees, our algorithm mitigates the risk of algorithmic complexity attacks (e.g., Denial-of-Service attacks triggered by carefully crafted input sequences), thereby enhancing the reliability and security of production systems.

We do not foresee any specific negative societal consequences directly stemming from this algorithmic improvement, beyond the general dual-use nature of efficient computing infrastructure.

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

# A. Properizing

## A.1. Semantics Discrepancy

As discussed by Berglund & van der Merwe (2023), Byte Pair Encoding (BPE) tokenization generally follows one of two distinct operational semantics:

1. **Standard BPE Semantics** (as defined in Section 3.1): This approach operates on a *fixed priority schedule*. It iterates through the merge rules from highest to lowest priority, applying each rule exhaustively across the sequence. Crucially, the process is linear with respect to the rule list: once a priority level is passed, the algorithm never revisits it, regardless of whether subsequent merges create new opportunities for higher-priority rules.

2. **SentencePiece Semantics** (Kudo & Richardson, 2018): This approach operates on a *dynamic priority queue*. At each step, it identifies the adjacent token pair with the **global maximum priority** in the current sequence. Notably, a merge operation generates at most two new adjacent pairs that may possess even higher priorities. SentencePiece will immediately execute either of the merges, effectively "growing" the token recursively and prioritizing these local expansions over the previous lower-priority global candidates.

To formalize this inconsistency, we adopt the terminology established by Berglund & van der Merwe (2023). An *improper dictionary* is defined as one containing merge rules that violate the sequential assumption of the standard BPE. Specifically, if a rule $r_l$ produces a token that creates the context for a strictly higher-priority rule $r_h$, the standard BPE will not execute $r_h$, since the algorithm has strictly proceeded past its priority level. In contrast, SentencePiece would immediately identify $r_h$ as the new global maximum.

To bridge this inconsistency, we propose a method to **properize** the dictionary: reassigning rule priorities via topological sorting so that the standard BPE's execution yields results identical to those of SentencePiece. The method is a sufficient construction for properization by reordering the priorities; other constructions may exist to achieve the same goal.

## A.2. Dictionary Filtering

In practice, BPE dictionaries often contain redundant or unreachable rules that are never triggered, even under SentencePiece semantics. To ensure the formalization is well-defined, we restrict our analysis to valid rules prior to properization.

Given the deterministic nature of BPE, we define a merge rule $(x, y) \to z$ as *valid* if and only if the tokenization of the concatenated string $xy$ under SentencePiece semantics results in the single token $z$ itself, and the final applied merge is indeed $(x, y)$. Rules failing this condition are effectively unreachable in any context and are thus discarded.

Crucially, this implies that the mapping from a produced token to its generating rule is unique; no two distinct valid rules result in the same token after filtering.

In the remainder of this section, we assume the dictionary has been filtered to contain solely valid rules. Based on this filtered dictionary, we can now define the dependency structure.

## A.3. Growing Tree

The SentencePiece semantics can be conceptualized as the standard BPE mechanism equipped with additional **growing steps**. Under this view, a merge operation $(x_0, y_0) \to z_0$ serves as a *seed event* that may trigger a cascade of immediate high-priority merges due to the greedy strategy.

Specifically, after the seed merge $(x_0, y_0)$, the new token $z_0$ immediately forms new pairs with its left neighbor $(u_1, z_0)$ and right neighbor $(z_0, v_1)$. If either of these new pairs has a strictly higher priority than the original seed $(x_0, y_0)$, SentencePiece will immediately execute the one with the higher priority. Note that executing one side implicitly invalidates the other, as the central token $z_0$ is consumed.

This recursive process, denoted as $(x_k, y_k) \to z_k$, repeats until no adjacent pair has a higher priority than the original seed $(x_0, y_0)$. The entire sequence behaves like "growing" outward from the initial seed merge. Note that, since the length of the merged token $z_k$ strictly increases with each step, no subsequent adjacent pair will possess the exact same priority as the seed.

Consider all possible scenarios of this process. The growing path from a seed extends either to the left or to the right,

forming a chain of growing steps. We formalize the collection of all such possible growing steps as the **Growing Tree**. Specifically, the root is the seed rule, and the child nodes are all valid merge rules that can be triggered by the parent during the growing of the root, distinguished by their growing direction (left or right).

To systematically construct these trees, we analyze the derivation of each valid rule to determine its role (root or child). For any string $s$, let $L(s)$ denote the lowest priority among all merge rules involved in the tokenization of $s$ under SentencePiece semantics. For a valid merge rule $r : (x, y) \to z$, it inherently holds that $L(z) \leq \min(L(x), L(y))$. We classify the rule $r$ based on this relationship:

1. **Root Rule** $(L(z) < \min(L(x), L(y)))$: The rule $r$ has a lower priority than any rule contained within its components. Thus, $r$ is a seed event that initiates a growing process. In the Growing Tree, $r$ is a root node. Functionally, we treat the root as *left-growing*, since the rule becomes applicable only after the token $y$ is generated.

2. **Growing Step** $(L(z) = \min(L(x), L(y)))$: The rule $r$ is triggered by a lower-priority bottleneck within its components. We determine the growing direction as follows:

    - **Left-Growing**: If $L(z) = L(y)$, the seed lies within $y$. Rule $r$ extends the growing chain to the left. **Crucially, this case covers the scenario where** $L(x) = L(y)$. Under SentencePiece semantics, such ties must be resolved as left-growing; otherwise, the necessary right-side token $y$ would not yet be established to participate in the merge. In the tree, $r$ is a child of the corresponding rule in $y$.
    - **Right-Growing**: If $L(z) = L(x)$ and $L(z) < L(y)$, the seed lies within $x$. Rule $r$ extends the growing chain to the right. In the tree, $r$ is a child of the corresponding rule in $x$.

Since the dictionary is filtered to ensure unique derivations, each merge rule corresponds to at most one parent, ensuring the dependency structure forms a valid forest.

Consequently, the growing process of a seed merge under SentencePiece semantics is equivalent to a greedy search for the deepest applicable path within the Growing Tree rooted at that seed.

### A.4. Dependency Graph and Topological Sort

To simulate SentencePiece semantics using the standard BPE, we need to flatten each Growing Tree into a valid sequential order, by reordering the priorities of rules during growing steps.

However, the growing steps are applied independently for each seed under SentencePiece semantics, while using the standard BPE requires each rule to be applied to the sequence exhaustively. We model the relative order and independence of growing steps as a dependency graph.

1. **Tree Edges (Parent $\to$ Child)**: Within each Growing Tree, a parent rule must be processed before its children. Although the children originally had higher priorities, in the properized dictionary, we must place them *after* the parent to ensure the parent exists to trigger them.

2. **Sibling Edges (Among the Direct Children)**: For siblings, during the growing steps, only the highest-prioritized applicable merge is selected. This implies that sibling rules are evaluated strictly in the order of their original priorities. Thus, we add edges from the siblings with higher priority to those with lower priority.

3. **Conflict Edges (Right-Growing $\to$ Left-Growing)**: A critical inconsistency arises when two growing steps from different trees compete for the same overlapping token. Suppose a *left-growing* rule $(w, x)$ and a *right-growing* rule $(y, w)$ both require token $w$.

    - Under SentencePiece semantics, each growing process runs independently, so the leftmost seed can obtain $w$ before others.
    - To emulate this deterministically without lookahead, we rely on the left-to-right scan nature of the standard BPE.
    - This imposes the constraint that **right-growing nodes must precede left-growing nodes**.

    Hence, edges from the right-growing node to the left-growing node obtaining the same token are required.

One can optimize the number of edges via transitive reduction. This simplifies the graph without altering its topological properties.

If the constructed dependency graph is a directed acyclic graph (DAG), a topological sort exists. Deriving the new rule priorities from this topological order yields a **properized dictionary**. The standard BPE, executing this dictionary linearly, will respect all dependencies of the growing steps and produce output identical to SentencePiece.

### A.5. Sufficiency

**Theorem A.1** (Sufficiency). *For any acyclic dependency graph, the derived properized dictionary yields tokenization results under the standard BPE that are identical to those under SentencePiece semantics.*

*Proof.* The proof establishes that a topological order exists to reconcile the execution discrepancy between the standard BPE and SentencePiece semantics.

- **Execution Discrepancy**: The standard BPE applies each merge rule globally and exhaustively across the entire sequence. In contrast, SentencePiece processes seeds sequentially from left to right, completing all recursive growing steps for the current seed before considering the next.

- **Intra-tree Consistency**: In the absence of conflicts between seeds with the same priority, the *Tree Edges* and *Sibling Edges* are sufficient to simulate SentencePiece's behavior. These edges ensure that the traversal of the deepest applicable path within each growing tree is executed in the correct dependency order.

- **Inter-tree Conflict Resolution**: When growing steps from different seeds of the same priority compete for overlapping tokens, SentencePiece inherently prioritizes the leftmost seed. To simulate this without lookahead, the *Conflict Edges* ensure that right-growing steps precede left-growing steps, effectively granting priority to the leftmost seeds within the global scan.

If the resulting dependency graph is a DAG, the topological sort provides a static priority order that ensures the standard BPE's global execution results in identical outputs.

Since *Tree Edges* and *Sibling Edges* define a directed forest, they are inherently acyclic. A cycle would only indicate a conflict between growing processes that cannot be resolved via static priority reordering. □

### A.6. Non-properizable scenarios

If the graph contains a cycle, the dictionary is **non-properizable**.

A canonical example is the dictionary `[(aa, a), (a, a)]` applied to the string "aaaa".

- **SentencePiece**: yields `[aaa, a]`.

- **The standard BPE**: yields `[aa, aa]`, no matter the priorities of the merge rules.

In the dependency graph, `(a, a)` acts as a root (left-growing) pointing to a right-growing node `(aa, a)`. However, the *Conflict Edges* would cause a dependency cycle, so it fails to properize.

In our experiments, one non-properizable tokenizer we found is Gemma-3 (Google, 2025b), which contains improper merge rules of multiple newline characters.

## B. Proof of the Prefix Consistency

In this section, we provide the formal proof for Lemma 3.1. Our proof relies on the inductive properties of the single merge operation visualized in Figure 4.

**Notation.** Let $\mathbb{T}_{(u,v)}$ denote the operator for a single BPE merge step that replaces all adjacent occurrences of tokens $(u, v)$ from left to right with the merged token $w$. Initially, the input string $s$ is mapped to a sequence of atomic tokens $\varphi_0$. The full BPE process is a composition of such operators: $\mathbb{T} = \mathbb{T}_{r_m} \circ \cdots \circ \mathbb{T}_{r_1}$, where $r_1, \ldots, r_m$ are the merge rules applied in order.

*Proof.* We prove the lemma by induction on the number of merge steps.

**Base Case (0 merges)**: The initial token sequence $\varphi_0$ consists of atomic tokens (bytes/characters). The boundaries are fixed at every character position. Trivially, any prefix of the atomic token sequence corresponds to the prefix of the raw text.

**Inductive Step**: Assume the consistency property holds for the state $\varphi_k$ after $k$ merges. Consider the $(k + 1)$-th merge rule $r = (u, v) \rightarrow w$. Let the state transition be $\varphi_{k+1} = \mathbb{T}_r(\varphi_k)$. Let $\mu$ be any prefix of the new token sequence $\varphi_{k+1}$.

We examine the **boundary** at the end of $\mu$. Since $\mathbb{T}_r$ is a **boundary elimination** operation, it only removes the internal boundaries between instances of $u$ and $v$ sequentially from left to right. It does not shift or create boundaries. Therefore, the boundary ending $\mu$ must have existed in the previous state $\varphi_k$. This implies there exists a unique prefix $\eta$ in $\varphi_k$ such that $\pi(\eta) = \pi(\mu)$.

Since $\mathbb{T}_r(\eta) = \mu$ (as the merge operation is local and respects the boundary), and $\eta$ is a valid tokenization by hypothesis, it follows that $\mu$ is the valid tokenization of $\pi(\mu)$ after the $(k + 1)$-th step. □

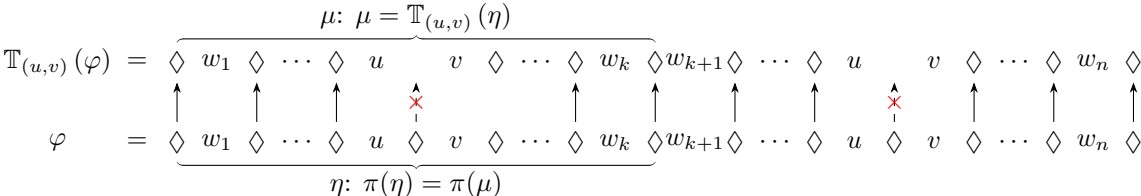

*Figure 4.* **Inductive step of the Prefix Consistency.** Solid/dashed arrows mark surviving/eliminated boundaries during a merge $(u, v)$. For any prefix $\mu$ in the post-merge token sequence, there exists exactly one valid prefix $\eta$ in the pre-merge token sequence with $\pi(\eta) = \pi(\mu)$. Notably, applying the single merge step to $\eta$ yields exactly $\mu = \mathbb{T}_{(u,v)}(\eta)$. This proves the lemma by induction over the full tokenization process.

## C. Normalization and Equivalence Details

In this section, we elaborate on the properties of the normalized representation introduced in Section 3.3.

**Canonical Generation and Closure.** For a canonical token $t$ produced eventually by the rule $(x, y) \rightarrow t$, the tokenization of the string $t$ must conclude with this specific merge. This necessitates that the substrings $x$ and $y$ must have been independently tokenized into $x$ and $y$ prior to this final merge. Therefore, $\mathbb{T}(x) = [x]$ and $\mathbb{T}(y) = [y]$, implying that the predecessor $\mathrm{pre}(t) = x$ and successor $\mathrm{suc}(t) = y$ are themselves canonical tokens. This ensures that $\mathcal{V}$ is closed under decomposition, which is essential for the construction of the Successor Forest.

**Equivalence.** Normalization does not alter tokenization results for general text. Since non-canonical rules are by definition never triggered in the deterministic BPE tokenization of valid tokens, removing them from the dictionary does not affect the merge process for any input string. Thus, $\mathbb{T}_{\mathcal{D}}(s) \equiv \mathbb{T}_D(s)$.

## D. Extended Intuition of Monotonic Path: Boundary Elimination

In this section, we expand on the intuition provided in Section 4.1 by reasoning about the merge process of the rightmost token.

Fix a non-empty string $s$ and a suffix token $t$ of $s$. Let $s^{-\mathrm{suc}(t)}$ denote the shorter prefix obtained by removing the suffix $\mathrm{suc}(t)$ from $s$. If $t$ is ever produced as the last token during the merge process, then immediately before applying the canonical rule of $t$, the tokens $\mathrm{pre}(t)$ and $\mathrm{suc}(t)$ must appear consecutively at the end of the current token sequence.

Crucially, at this specific moment, the boundary between $\mathrm{pre}(t)$ and $\mathrm{suc}(t)$ remains intact, so the presence of the rightmost token $\mathrm{suc}(t)$ does not affect any merges that have already occurred to the left of $\mathrm{pre}(t)$. Therefore, the token sequence obtained by removing the final $\mathrm{suc}(t)$ coincides with the token sequence produced by the merge process of $s^{-\,\mathrm{suc}(t)}$ right before applying the rule of $t$.

This implies a necessary condition: if $t$ is to be formed, $\mathrm{pre}(t)$ must appear as the rightmost token at some stage during the merge process of $s^{-\,\mathrm{suc}(t)}$. In terms of the Successor Forest structure, this means that the eventual last token $\theta(s^{-\,\mathrm{suc}(t)})$ must be a descendant of $\mathrm{pre}(t)$ (or $\mathrm{pre}(t)$ itself).

Additionally, whether this state is reachable depends on rule priorities. If $\mathrm{pre}(t)$ is not yet the final result $\theta(s^{-\,\mathrm{suc}(t)})$, it is destined to be merged with a left neighbor to form a larger token (specifically, one of its children in the Successor Forest, formed by merging it with its left neighbor). For $t$ to exist, the canonical rule of $t$ must be applied *before* this specific merge event occurs.

In summary, the formation of $t$ is determined by a single critical moment: whether $\mathrm{pre}(t)$ can persist as the rightmost token of $s^{-\,\mathrm{suc}(t)}$ until the canonical rule of $t$ is applied.

## E. Proof of Monotonic Path Property

In this section, we prove Theorem 4.2, which characterizes the structure of suffix tokens that satisfy the Prefix Last-Token Condition (Definition 4.1).

Recall that for a non-empty string $s$, let $k$ be the longest canonical suffix token of $s$, and let $\mathrm{SufSucTree}(k)$ denote the Suffix-Successor Tree induced by all canonical suffix tokens of $k$. The theorem asserts that the set of tokens in $\mathrm{SufSucTree}(k)$ satisfying the Prefix Last-Token Condition forms exactly the unique path from the true last token $\theta(s)$ to the root of $\mathrm{SufSucTree}(k)$.

### E.1. Proof Overview

As discussed in Section 4.1 and Appendix D, the Prefix Last-Token Condition (Definition 4.1) can be intuitively understood as the ability of a node $w$ to *"steal"* its predecessor $\mathrm{pre}(w)$ from the last token $\theta(s^{-\,\mathrm{suc}(w)})$ of the truncated string.

Claim 1 establishes an *upward-closure* property: if a node has this stealing capability, then its successor must also have it. Otherwise, the canonicity of the involved tokens would be violated.

Claim 2 addresses the question of uniqueness: whether multiple children of the same successor could simultaneously satisfy the condition. The determinism of the standard BPE merge process, together with strict priority ordering, rules out this possibility.

Claim 3 confirms that the true last token $\theta(s)$ indeed satisfies the condition. Intuitively, since $\theta(s)$ is produced by the final merge, it necessarily has the ability to steal $\mathrm{pre}(\theta(s))$.

Finally, Claim 4 shows that the answer node $\theta(s)$ cannot be further extended: if it could steal once more, it would not be the last token of the final tokenization of the string.

With these claims, the Monotonic Path Property follows.

### E.2. Detailed Proof

We begin by introducing auxiliary notation and several structural lemmas that will be used throughout the proof.

**Ancestor Path.** For any canonical token $x \in \mathcal{V}$, we define $P(x)$ to be the unique path from $x$ to the root of the Successor Forest $\mathcal{F}_{\mathrm{suc}}$, following successor edges. Equivalently, $P(x)$ consists of $x$ and all of its ancestors in the Successor Forest.

Since the Successor Forest is acyclic and every non-atomic token has a unique successor, $P(x)$ is well-defined and totally ordered by ancestry.

**Rule Priority Convention.** Throughout the proof, we compare merge rules by their execution order in the dictionary: a rule $r_i$ has higher priority than $r_j$ (i.e., $r_i > r_j$) if and only if $r_i$ is applied earlier in the standard BPE merge process.

**Lemma E.1** (Priority Monotonicity Along Successor Paths). *Let $x, y \in \mathcal{V}$ be canonical tokens such that $y$ is a descendant of $x$ in the Successor Forest (i.e., $x \in P(y)$). If the token $x$ is non-atomic, then the canonical merge rule producing the token $x$ has strictly higher priority than the canonical merge rule producing the token $y$.*

*Proof.* By definition of the Successor Forest, $y$ is obtained by a sequence of canonical merges starting from $x$. If the canonical rule producing $y$ were applied before the canonical rule producing $x$, then $x$ would never appear as an isolated token during the tokenization of its own string, contradicting the assumption that $x$ is canonical. □

**Lemma E.2** (Shared Right Boundary). *Let $x, y \in \mathcal{V}$ be canonical tokens such that $x \in P(y)$. Consider any tokenization process in which the token $y$ appears as a token in the token sequence. Immediately before the canonical rule producing the token $y$ is applied, the token $x$ must also appear as a token, and the right boundary of $x$ coincides with the right boundary of $y$.*

*Proof.* Since $y$ is produced by successive canonical merges starting from $x$, all intermediate merges preserve the rightmost boundary until the final merge forming $y$. Thus, prior to the execution of the canonical rule of $y$, the token $x$ must exist and share the same right endpoint. □

**Lemma E.3** (Ancestor Characterization of Last Tokens). *For any non-empty string $s$ and any canonical token $x$, we have*

$$x \in P(\theta(s)) \iff x \text{ appears as the last token at some stage during the tokenization of the string } s.$$

*Proof.* ($\Rightarrow$) If $x \in P(\theta(s))$, then $\theta(s)$ is obtained from $x$ by a sequence of canonical merges. Immediately before each such merge, $x$ (or its descendant) appears as the last token.

($\Leftarrow$) If $x$ appears as the last token at some stage, then the last token $\theta(s)$ must be obtained by further merges starting from $x$, implying $x \in P(\theta(s))$. □

**Lemma E.4** (Tokenization Consistency of Token-wise Truncation). *The Prefix Consistency (Lemma 3.1) extends naturally to any contiguous subsequence of the token sequence during the tokenization. It also applies to any prefix of merge rules in a proper dictionary.*

*Specifically, for any integer $k \leq |\mathcal{D}|$, after applying the first $k$ merge rules to a string and obtaining the token sequence $\varphi$, if we perform token-wise truncation on $\varphi$, the isolated contiguous subsequence $\mu$ is exactly the valid tokenization of its corresponding underlying string $\pi(\mu)$ with the first $k$ merge rules.*

*Proof.* This follows directly from the inductive proof of Lemma 3.1, which applies to each individual merge step.

From a more abstract perspective, because the boundaries delimiting $\mu$ survive the first $k$ merges, no merge operation has crossed them. Due to the deterministic, left-to-right, non-overlapping nature of BPE tokenization, the sequence of internal merges strictly within $\mu$ is entirely independent of the tokens outside these boundaries. Therefore, isolating $\mu$ via token-wise truncation yields the exact same state as independently tokenizing the raw string $\pi(\mu)$ with the first $k$ rules. □

**Lemma E.5** (Step-wise Consistency of Token-wise Truncation). *Let $s$ be a string and $\varphi$ be its token sequence after applying the first $k$ merge rules. Let $\mu$ be a contiguous subsequence of $\varphi$ obtained by token-wise truncation. Let $r$ be the $(k+1)$-th merge rule. Applying $r$ to $\mu$ (i.e., $\mathbb{T}_r(\mu)$) yields exactly the valid tokenization of the corresponding substring $\pi(\mu)$ with the first $k+1$ merge rules.*

*Proof.* By Lemma E.4, token-wise truncation is valid after $k$ merges. Thus, $\mu$ is exactly the correct token sequence for the substring $\pi(\mu)$ after the first $k$ rules.

By the definition of the BPE tokenization, the tokenization after $k+1$ rules is obtained by applying the $(k+1)$-th rule $r$ directly to the tokenization resulting from the first $k$ rules.

Therefore, $\mathbb{T}_r(\mu)$ yields the tokenization of $\pi(\mu)$ after applying the first $k+1$ merge rules. □

**Lemma E.6** (Exclusive Priorities of Tokens satisfying the Prefix Last-Token Condition). *Let $t$ be a non-atomic suffix token of a string $s$ that satisfies the Prefix Last-Token Condition (Definition 4.1). During the tokenization process of $s$, immediately prior to executing the canonical rule of $t$, the rightmost two tokens in the sequence are exactly $\mathrm{pre}(t)$ and $\mathrm{suc}(t)$.*

*Proof.* For the string $s$, let the token boundary immediately to the left of the suffix string $\mathrm{suc}(t)$ be $\Diamond_{\mathrm{suc}(t)}$, the token boundary immediately to the left of the suffix string $t$ be $\Diamond_t$. We term the token boundaries strictly between $\Diamond_t$ and $\Diamond_{\mathrm{suc}(t)}$ as the *internal boundaries* of $\mathrm{pre}(t)$.

The following proof will focus on the status of $\Diamond_t$, $\Diamond_{\mathrm{suc}(t)}$, and the internal boundaries of $\mathrm{pre}(t)$.

**Step 1: The internal boundaries of $\mathrm{pre}(t)$ must be eliminated first.** We first prove that during the merges before applying the canonical rule of $t$, no rule can eliminate $\Diamond_t$ or $\Diamond_{\mathrm{suc}(t)}$ before all token boundaries between them are eliminated.

Suppose a merge rule $r$ attempts to eliminate one or more of these boundaries while at least one internal boundary still exists:

- If the rule $r$ eliminates $\Diamond_t$: By Lemma E.5, we can perform token-wise truncation at $\Diamond_{\mathrm{suc}(t)}$ immediately before the rule $r$, leaving the tokenization of the prefix string $s^{-\,\mathrm{suc}(t)}$. The tokenization before and after the rule $r$ must be valid for $s^{-\,\mathrm{suc}(t)}$. However, with the merge rule $r$, $\Diamond_t$ is eliminated before the internal boundaries of $\mathrm{pre}(t)$ are cleared, $\mathrm{pre}(t)$ can never become an isolated token as the last token during the tokenization. This leads to $\mathrm{pre}(t) \notin P(\theta(s^{-\,\mathrm{suc}(t)}))$, which contradicts the **Reachability** condition.

- If the rule $r$ eliminates $\Diamond_{\mathrm{suc}(t)}$ while $\Diamond_t$ survives: We truncate at $\Diamond_t$. The suffix should form the valid tokenization of $t$. However, the boundary between $\mathrm{pre}(t)$ and $\mathrm{suc}(t)$ is eliminated before the internal boundaries of $\mathrm{pre}(t)$ are cleared. This contradicts the proper merge sequence of the canonical token $t$.

Furthermore, if the internal boundaries exist, there are at least 3 consecutive token boundaries. Since BPE only merges adjacent pairs without overlapping, the rule $r$ can eliminate at most half of the consecutive boundaries. Thus, it is impossible for all of $\Diamond_t$, $\Diamond_{\mathrm{suc}(t)}$, and the internal boundaries of $\mathrm{pre}(t)$ to be eliminated simultaneously.

This confirms that any attempt to eliminate $\Diamond_t$ or $\Diamond_{\mathrm{suc}(t)}$ must happen strictly after the internal boundaries are fully eliminated.

Furthermore, since the truncation between $\Diamond_t$ and $\Diamond_{\mathrm{suc}(t)}$ yields the valid tokenization of $\mathrm{pre}(t)$, and the canonical rule of $\mathrm{pre}(t)$ has a strictly higher priority than the canonical rule of $t$ (or $\mathrm{pre}(t)$ is an atomic token), it follows that immediately before executing the canonical rule of $t$ during the tokenization of $s$, both $\Diamond_t$ and $\Diamond_{\mathrm{suc}(t)}$ must survive without any internal boundaries between them, which means $\mathrm{pre}(t)$ between $\Diamond_t$ and $\Diamond_{\mathrm{suc}(t)}$ should become a single isolated token.

**Step 2: $\mathrm{pre}(t)$ cannot be consumed before the canonical rule of $t$ is executed.** Next, consider the process after $\mathrm{pre}(t)$ is completely formed but before the canonical rule of $t$ is executed.

- If $\mathrm{pre}(t)$ merges leftward (acting as a successor): The boundary $\Diamond_t$ is eliminated. Since a single BPE merge eliminates at most one boundary around each token, $\Diamond_{\mathrm{suc}(t)}$ must survive. Truncating at $\Diamond_{\mathrm{suc}(t)}$, the prefix corresponds to the tokenization of $s^{-\,\mathrm{suc}(t)}$. This merge implies $\mathrm{pre}(t)$ is merged into a larger token during the tokenization of $s^{-\,\mathrm{suc}(t)}$, so $\theta(s^{-\,\mathrm{suc}(t)}) \neq \mathrm{pre}(t)$. By definition, the executed rule is exactly the unique child of $\mathrm{pre}(t)$ on $P(\theta(s^{-\,\mathrm{suc}(t)}))$. However, the **Priority dominance** requires the canonical rule of $t$ to have a strictly higher priority than this child. Thus, it is impossible for this merge to occur before the canonical rule of $t$, contradicting the assumption.

- If $\mathrm{pre}(t)$ merges rightward: The merge eliminates $\Diamond_{\mathrm{suc}(t)}$ while $\Diamond_t$ survives. Truncating at $\Diamond_t$, the suffix must represent the valid tokenization of $t$. However, the canonical rule of $t$ is not yet applied, while the prefix token $\mathrm{pre}(t)$ merges rightward with a token other than $\mathrm{suc}(t)$. This contradicts the proper merge sequence of the canonical token $t$.

**Step 3: The right side must solely be $\mathrm{suc}(t)$.** Finally, immediately prior to executing the canonical rule of $t$, suppose there are still other surviving token boundaries between $\Diamond_{\mathrm{suc}(t)}$ and the end of the sequence. If we truncate at $\Diamond_t$, the isolated suffix sequence would contain more than two tokens. However, in the valid tokenization of the canonical token $t$, the token sequence exactly prior to applying its canonical rule must consist of exactly two tokens: $\mathrm{pre}(t)$ and $\mathrm{suc}(t)$. Contradiction.

**Conclusion.** Thus, immediately prior to executing the canonical rule of $t$, the token sequence ends exactly with $\mathrm{pre}(t)$ and $\mathrm{suc}(t)$. $\qquad\square$

With the above notation and lemmas in place, we now prove Theorem 4.2 by establishing four structural claims: upward closure, uniqueness, validity of $\theta(s)$, and maximality.

**Claim 1 (Upward Closure).** Let $s$ be a non-empty string and let $k$ be the longest canonical suffix token of $s$. For any node $w \in \mathrm{SufSucTree}(k)$, if $w$ satisfies the Prefix Last-Token Condition (Definition 4.1) with respect to $s$, then its parent $\mathrm{suc}(w)$ in the Successor Forest (if it exists) also satisfies the Prefix Last-Token Condition.

*Proof.* If $w$ is an atomic token, it is a root, making the claim vacuous. Assume $w$ is non-atomic. Let $v = \mathrm{suc}(w)$. We aim to prove $v$ satisfies the condition. If $v$ is an atomic token, it satisfies the condition by definition. Thus, we further assume $v$ is non-atomic.

**Notations.** We define the following components and their corresponding canonical merge rules:

- $u = \mathrm{pre}(w)$ and $v = \mathrm{suc}(w)$, with $r_w$ being the canonical rule $(u, v) \to w$.

- $x = \mathrm{pre}(v)$ and $y = \mathrm{suc}(v)$, with $r_v$ being the canonical rule $(x, y) \to v$.

Since $w$ is a suffix of $s$ and $v$ is a suffix of $w$, the token $y$ is exactly the suffix token of $s$. Let $s^{-y}$ denote the prefix string obtained by removing $y$ from $s$.

**Step 1: Reachability.** Because $w$ satisfies the condition, Lemma E.6 guarantees that during the tokenization of $s$, immediately prior to executing $r_w$, the rightmost two tokens are exactly $u$ and $v$. The existence of the fully formed token $v$ requires that its canonical rule $r_v$ must have been executed at some earlier stage.

Backtracking to the state immediately prior to the execution of $r_v$, the token sequence must end exactly with $x$ and $y$. At this specific moment, the token boundary between $x$ and $y$ exists. By Lemma E.5, performing token-wise truncation at this boundary (removing $y$) yields the valid tokenization of $s^{-y}$ at this stage, ending with the token $x$. By Lemma E.3, since $x$ appears as the last token during the tokenization of $s^{-y}$, we have $x \in P(\theta(s^{-y}))$. This establishes the **Reachability** condition for $v$.

**Step 2: Priority Dominance.** Suppose $x \neq \theta(s^{-y})$. This implies the final $x$ of $s^{-y}$ will eventually be consumed leftward by some merge rule $r_{c_x}$ to form $c_x$ (the unique child of $x$ on $P(\theta(s^{-y}))$).

Let $\varphi$ be the token sequence of $s$ immediately prior to executing $r_v$. As established in Step 1, $\varphi$ ends exactly with $x$ and $y$, so we can write $\varphi = \eta \oplus [x, y]$ for some prefix sequence $\eta$. By Lemma E.5, performing token-wise truncation yields $\mu = \eta \oplus [x]$, which is exactly the token sequence of $s^{-y}$ at this same stage.

Since the final token $x$ remains unmerged in $\mu$, the rule $r_{c_x}$ has not yet been applied. Thus, $r_{c_x}$ cannot have a higher priority than $r_v$.

Suppose their priorities are equal, meaning $r_{c_x}$ is exactly $r_v$, i.e., $c_x = v$. For $r_v$ to consume $x$ leftward, given its definition $(x, y) \to v$, we must have $x = y$. This makes the rule $r_v = (x, x) \to v$. As $r_{c_x}$ must be applied and alter the last token of $\mathbb{T}_{r_{c_x}}(\mu)$ to $v$, the token sequences can be written as $\mu = \eta' \oplus [x, x]$ and $\varphi = \eta' \oplus [x, x, x]$. Now consider the deterministic left-to-right matching of the rule $(x, x) \to v$. Either $\mathbb{T}_{r_v}(\mu)$ or $\mathbb{T}_{r_v}(\varphi)$ exactly ends with the token $v$, while the other ends with the token $x$, contradicting the analysis where both must end with $v$.

Therefore, the canonical rule $r_v$ must have a strictly higher priority than $r_{c_x}$, satisfying the **Priority dominance** condition for token $v$ with respect to $s$.

**Conclusion.** With both conditions verified, $\mathrm{suc}(w) = v$ satisfies the Prefix Last-Token Condition with respect to $s$. $\square$

**Claim 2 (Uniqueness of the satisfying child).** Let $s$ be a non-empty string. For any node $v$, there exists *at most one* child $w$ of $v$ that satisfies the Prefix Last-Token Condition (Definition 4.1) with respect to $s$.

*Proof.* Assume for contradiction that there exist two distinct nodes $w_a \neq w_b$ that both satisfy the condition with respect to $s$, and share the same successor $v = \mathrm{suc}(w_a) = \mathrm{suc}(w_b)$. Let $u_a = \mathrm{pre}(w_a)$ and $u_b = \mathrm{pre}(w_b)$. Let $r_{w_a} = (u_a, v)$ and $r_{w_b} = (u_b, v)$. Since $w_a \neq w_b$, we have $u_a \neq u_b$.

By the **Reachability** condition, both $u_a, u_b \in P(\theta(s^{-v}))$. Since $P(\theta(s^{-v}))$ is a chain on a tree, one must be an ancestor of the other. Without loss of generality, assume $u_a$ is a proper ancestor of $u_b$. Thus $u_a \neq \theta(s^{-v})$.

This implies that during the tokenization of $s^{-v}$, the token $u_a$ is formed earlier than $u_b$ and is required to form $u_b$. For $u_b$ to eventually form, $u_a$ must uniquely merge leftward via some rule $r_{c_a} = (k, u_a) \to c_a$, where $c_a$ is the unique child of $u_a$ on $P(u_b)$. Since $u_a, u_b \in P(\theta(s^{-v}))$, $c_a$ is also the unique child of $u_a$ on $P(\theta(s^{-v}))$.

By the **Priority dominance** condition for $w_a$, we have $r_{w_a} > r_{c_a}$ and $w_a \neq c_a$.

Because $w_a$ satisfies the condition, Lemma E.6 guarantees that during the tokenization of $s$, immediately prior to executing the rule $r_{w_a}$, the rightmost two tokens are exactly $u_a$ and $v$. Likewise, immediately prior to executing the rule $r_{w_b}$, the rightmost two tokens are exactly $u_b$ and $v$.

Because $u_a$ is a proper ancestor of $u_b$, during the forward tokenization of $s$, the state where $u_a$ and $v$ are the rightmost two tokens must strictly precede the state where $u_b$ and $v$ are the rightmost two tokens.

During the tokenization of $s$, when executing the rule $r_{w_a}$, the token boundary immediately to the left of the last token $v$ must survive the merge; otherwise, the last token would never be $v$ immediately prior to executing the rule $r_{w_b}$ later on. This means that the execution of $r_{w_a}$ leaves a mergeable adjacent token pair $[u_a, v]$ intact at the end. However, by the left-to-right nature of BPE, this will only occur when the predecessor and the successor of the merge rule are the same, and there is a sequence of at least three such identical tokens being adjacent. **Thus, we must have $u_a = v$ and $w_a$ is a child of $u_a$.**

Hence, immediately prior to executing the rule $r_{w_a}$ during the tokenization of $s$, the rightmost three tokens in the sequence are exactly the same: the token $v$. After executing the rule $r_{w_a}$, the rightmost two tokens become exactly $w_a$ and $v$. Consider the token-wise truncation removing the last token $v$ to obtain the valid tokenization of $s^{-v}$. At this specific stage, we have $w_a$ as the last token of the tokenization of $s^{-v}$, i.e., $w_a \in P(\theta(s^{-v}))$.

Since $w_a$ is a child of $u_a$, $w_a$ is the unique child of $u_a$ on $P(\theta(s^{-v}))$, contradicting $w_a \neq c_a$.

Therefore, the assumption is false. $\qquad\square$

**Claim 3 (The answer node satisfies the condition).** Let $w = \theta(s)$ be the last token for a non-empty string $s$ after tokenization. Then $w$ satisfies the Prefix Last-Token Condition (Definition 4.1) with respect to $s$.

*Proof.* If $w$ is an atomic token, it satisfies the condition by definition. Assume $w$ is non-atomic. Let $u = \mathrm{pre}(w)$ and $v = \mathrm{suc}(w)$, with $r_w$ being the canonical merge rule $(u, v) \to w$.

Since $w = \theta(s)$, the tokenization of $s$ eventually produces $w$ as its last token. This implies that immediately prior to the execution of $r_w$, the token sequence of $s$ must end exactly with $u$ and $v$.

**Step 1: Reachability.** At this exact moment, the boundary between $u$ and $v$ exists. By Lemma E.5, performing token-wise truncation at this boundary (removing $v$) yields the valid tokenization of $s^{-v}$ at this stage, which ends with the token $u$. By Lemma E.3, since $u$ appears as the last token during the tokenization of $s^{-v}$, we have $u \in P(\theta(s^{-v}))$. This establishes the **Reachability** condition for $w$.

**Step 2: Priority Dominance.** Suppose $u \neq \theta(s^{-v})$. Then the last token $u$ during the tokenization of $s^{-v}$ must eventually be consumed leftward by some merge rule $r_{c_u} = (k, u) \to c_u$, where $c_u$ is the unique child of $u$ on $P(\theta(s^{-v}))$.

Since $u$ appears as the last token, during the tokenization of $s^{-v}$ immediately prior to the execution of $r_w$, the rule $r_{c_u}$ has not yet been applied, meaning its priority cannot be strictly higher than $r_w$, i.e., $r_{c_u} \leq r_w$.

Suppose their priorities are equal, i.e., $c_u = w$. Since $r_w = (u, v)$ and $r_{c_u} = (k, u)$, we must have $u = v$, making the rule $r_w = (u, u) \to w$.

Since $u \neq \theta(s^{-v})$ and $c_u$ is the unique child of $u$ on $P(\theta(s^{-v}))$, during the tokenization of $s^{-v}$ when applying $r_{c_u} = r_w$, the last token must be altered to $w$ in order to form $\theta(s^{-v})$. However, since $w = \theta(s)$, during the tokenization of $s$ when applying $r_w$, the last token must be altered to $w$. Similar to the proof of Step 2 in Claim 1, this will be impossible. Thus $c_u \neq w$ and $r_{c_u} < r_w$.

This establishes the **Priority Dominance** condition for $w$. $\qquad\square$

**Claim 4 (The answer node has no children satisfying the condition).** Let $s$ be a non-empty string. The answer node $\theta(s)$ has no children that satisfy the Prefix Last-Token Condition (Definition 4.1) with respect to $s$.

*Proof.* Let $v = \theta(s)$. Assume for contradiction that there exists a child node $w$ of $v$ (meaning $v = \mathrm{suc}(w)$) that satisfies the condition with respect to $s$. Let $u = \mathrm{pre}(w)$, and let $r_w$ be the canonical merge rule $(u, v) \to w$.

Because $v = \theta(s)$, the Prefix Consistency (Lemma 3.1) guarantees that the final tokenization of $s$ is exactly the tokenization of $s^{-v}$ appended with $v$, i.e., $\mathbb{T}(s) = \mathbb{T}(s^{-v}) \oplus [v]$.

By the **Reachability** condition for $w$, we have $u \in P(\theta(s^{-v}))$. We consider the two possible cases for $u$:

**Case 1: $u = \theta(s^{-v})$.** If $u$ is exactly the last token of $s^{-v}$, then $\mathbb{T}(s^{-v})$ ends with $u$. Consequently, the final tokenization $\mathbb{T}(s)$ must end exactly with the adjacent pair of tokens $[u, v]$. However, since $r_w = (u, v) \to w$ is a valid merge rule in the vocabulary, the deterministic BPE process would not terminate leaving the matchable pair $[u, v]$ intact. If $u \neq v$, the tokenization of $s$ will not terminate with the isolated token $v$ as the last token. Even if $u = v$, in the final tokenization $\mathbb{T}(s)$, the rightmost two tokens would at most be $w$ and $v$, rather than two adjacent $u$'s, meaning the tokenization $\mathbb{T}(s^{-v})$ would have $w$ as the last token. Hence, $u$ will not be $\theta(s^{-v})$.

**Case 2: $u \neq \theta(s^{-v})$.** If $u$ is not the last token of $s^{-v}$, it must eventually be consumed leftward by some merge rule $r_{c_u} = (k, u) \to c_u$, where $c_u$ is the unique child of $u$ on $P(\theta(s^{-v}))$. Because $w$ satisfies the condition, the **Priority dominance** strictly dictates that $r_w > r_{c_u}$.

During the tokenization of the string $s$, since $v = \theta(s)$, the suffix token $v$ must never be consumed by any rightward merges after its formation. So the token boundary immediately to the left of the suffix $v$ is never eliminated. By Lemma E.4, we can truncate the tokenization of $s$ removing the suffix corresponding to the string $v$ and obtain the valid tokenization of $s^{-v}$ after the execution of each merge rule.

Consider the tokenizations of both $s^{-v}$ and $s$ immediately prior to executing $r_w$, where the tokens $u$ and $v$ have been formed, as they are the predecessor and the successor of $w$. At this stage, $r_{c_u}$ has not been executed since $r_w > r_{c_u}$. Hence, at this moment, the last token of the tokenization of $s^{-v}$ is $u$, and the last token of the tokenization of $s$ is $v$. So the rightmost two tokens of the tokenization of $s$ are $u$ and $v$.

However, since $v = \theta(s)$, the execution of $r_w$ must skip the rightmost adjacent $u$ and $v$, which is only possible when $u = v$. Similar to the proof of Step 2 in Claim 1, this will also be impossible, as it leads to the case where $w = c_u$, directly contradicting the requirement of **Priority dominance** ($r_w > r_{c_u}$).

**Conclusion.** Both cases yield a direct contradiction. Therefore, the assumption is false, and the answer node $\theta(s)$ has no children satisfying the condition. □

**Conclusion of the Proof.** We now conclude the proof of Theorem 4.2. Claim 1 establishes that the Prefix Last-Token Condition is upward closed along successor edges. Claim 2 shows that for any node, at most one child can satisfy the condition. Claim 3 guarantees that the true last token $\theta(s)$ does satisfy the condition, while Claim 4 shows that it is maximal and admits no further extension.

Together, these four claims imply that the set of tokens in $\mathrm{SufSucTree}(t)$ satisfying the Prefix Last-Token Condition forms exactly a single, maximal path from $\theta(s)$ to the root of the tree. This completes the proof.

# F. Square-Root Tiled Transition Table

In an Aho–Corasick automaton (Aho & Corasick, 1975), a straightforward implementation stores a full transition array over the alphabet $\Sigma$ for every state, yielding constant-time transitions at the cost of a large memory footprint.

A key observation is that, for any state, its transition table differs from that of its suffix link only at entries corresponding to outgoing trie edges. This structural locality allows transitions to be shared persistently across states.

We exploit this property using a square-root tiling of the alphabet. The alphabet is partitioned into fixed-size tiles; each state stores references to its tiles, and a tile is duplicated only when a transition within it is modified. As a result, most tiles are shared between a state and its suffix link, while updates remain local to the affected tiles.

This representation preserves $\mathcal{O}(1)$ transition queries and supports efficient construction, while substantially reducing memory usage in practice.

## G. Eager Output Algorithm Details

Based on the Active Frontier definition in Section 6.1, we maintain the subgraph of the Prefix Tree of Tokens composed of the nodes in $\mathcal{P}$ and their ancestors to identify the common ancestral path. The algorithm proceeds as follows:

1. **Window Maintenance**: We maintain $\mathcal{P}$ using two pointers. The right pointer adds the new $\theta(s)$. The left pointer advances to satisfy the bound $|s| - d(s)$, removing expired tokens, where the function $d$ is defined in Section 6.1.

2. **Subgraph Tracking**: We maintain the nodes in $\mathcal{P}$ and their ancestors as a dynamic subgraph. For each node in this subgraph, we record the number of its children that are also part of the subgraph.

3. **Eager Emission**: We maintain the children of the virtual root. If the virtual root has exactly **one child** $c$ in the subgraph, it implies that all Parental Candidates (and thus all possible futures) are descendants of the node $c$. The token represented by $c$ is therefore stable. We emit the corresponding token, update the virtual root to $c$, and repeat the check.

Each node in the Prefix Tree of Tokens is inserted into the maintained subgraph exactly once, at the moment when the right pointer includes the corresponding last token into $\mathcal{P}$. A node may persist in the subgraph even after it ceases to belong to $\mathcal{P}$, but it will be removed at most once, when it no longer lies on the ancestral path of any active Parental Candidate.

During the advancement of the left pointer, each step removes either one node from the subgraph or performs a constant amount of bookkeeping without removal. Consequently, each node contributes $\mathcal{O}(1)$ work over the entire execution of the algorithm.

Therefore, the eager output mechanism incurs an amortized $\mathcal{O}(1)$ time overhead per input update.

## H. Detailed Benchmarks

### H.1. Experiment Setup

All benchmarks were conducted on a bare-metal server equipped with an **Intel® Xeon® Platinum 8362 CPU @ 2.80GHz**. To ensure reproducibility and minimize measurement noise, we applied the following configurations:

- **CPU Isolation**: Experiments were pinned to a single NUMA node (Node 1) containing 32 physical cores and 128 GB of RAM.

- **System Tuning**: Hyper-threading (SMT), Turbo Boost, and Address Space Layout Randomization (ASLR) were disabled.

- **Concurrency**: Tests were executed using 30 parallel workers. Each worker ran as an isolated, single-threaded process that independently loaded the tokenizer and dataset.

To ensure the validity of our results, we verified the correctness of our implementations against baseline methods prior to performance benchmarking. For all benchmarks (`tokenizers` and `tiktoken` bindings), we changed the global allocator to `mimalloc` (Leijen et al., 2019; Purplecoin, 2026) and fixed seeds for hash functions to guarantee stable and reproducible latency measurements.

### H.2. Datasets

We constructed three representative datasets using specific subsets of publicly available datasets. Table 2 provides a summary of the dataset statistics.

The specific sampling strategies were as follows:

- **English & Chinese**: We utilized the first Parquet shard from the standard training split of the Wikipedia dataset (20231101 dump), and employed a strided sampling strategy, extracting every 42nd document for English and every 60th document for Chinese.

*Table 2.* **Datasets used in experiments.**

| Name (Reference) | Repository | #Documents | Total #Bytes |
|---|---|---|---|
| English (Wikimedia Foundation, 2023a) | Wikimedia Foundation, 2023b | 3,722 | 16,522,972 |
| Chinese (Wikimedia Foundation, 2023a) | Wikimedia Foundation, 2023b | 3,847 | 16,337,583 |
| Code (Together Computer, 2023a) | Together Computer, 2023b | 2,500 | 16,803,408 |

- **Code**: We extracted data from the RedPajama GitHub subset (Together Computer, 2023b), specifically using the file with the filename "`filtered_08cdfa755e6d4d89b673d5bd1acee5f6.sampled.jsonl`". We selected the first 2,500 documents from this file.

Additionally, for pathological input analysis, we constructed strings consisting of $2^k$ repetitions of the character 'a'.

### H.3. Tokenizers

We evaluated a diverse set of tokenizers supported by two major libraries: Hugging Face's `tokenizers` and OpenAI's `tiktoken`.

*Table 3.* **Tokenizers used in experiments for Hugging Face's `tokenizers`.**

| Name (Reference) | Repository | Vocab Size | P99 Token Length |
|---|---|---|---|
| CodeLlama (Meta AI, 2023) | Meta Llama, 2023 | 32,016 | 12 |
| DeepSeek-3.2 (DeepSeek-AI, 2025a) | DeepSeek-AI, 2025b | 128,000 | 15 |
| Gemma-3[†] (Google, 2025a) | Google, 2025b | 262,144 | 14 |
| GPT-OSS (OpenAI, 2025a) | OpenAI, 2025b | 199,998 | 19 |
| Llama-3.1[*] (Meta AI, 2024) | Meta Llama, 2024 | 128,000 | 17 |
| Llama-4 (Meta AI, 2025) | Meta Llama, 2025 | 200,000 | 21 |
| Mistral-3 (Mistral AI, 2025a) | Mistral AI, 2025b | 131,072 | 16 |
| Ouro (ByteDance, 2025a) | ByteDance, 2025b | 49,152 | 14 |
| Qwen-3 (Qwen Team, 2025a) | Qwen Team, 2025b | 151,643 | 18 |

[†] Improper dictionary. See Appendix A.6 for non-properizable cases.  [*] Properized dictionary. See Appendix A for properization.

**Hugging Face Models.** Table 3 summarizes the open-source tokenizers selected for our experiments. These models exhibit varying pre-tokenization strategies:

- **Regex-Based**: The majority of the models (DeepSeek-3.2, GPT-OSS, Llama-3.1, Llama-4, Mistral-3, and Qwen-3) employ regex-based rules to split input text before tokenization.

- **Rule-Based**: Ouro utilizes a digit-based pre-tokenization, and Gemma-3 employs a rule on space characters.

- **None**: Notably, CodeLlama applies **no pre-tokenization** logic.

Regarding the "properness" of dictionaries, the improper merge rules of Llama-3.1 are properized for our benchmarks (see Appendix A). In contrast, Gemma-3's merge rules are improper and cannot be properized (see Appendix A.6).

**OpenAI Models.** For the `tiktoken` library, we utilized the standard built-in encoding models. Table 4 lists the specific models that we used. Note that all tested `tiktoken` models employ regex-based pre-tokenization.

### H.4. Detailed Results

We report throughput in terms of MiB/s based on the median of multiple execution runs. For each benchmark setting, throughput is computed as the total number of input bytes divided by the sum of the per-document median execution times. This aggregation reflects the effective end-to-end processing rate under repeated, independent tokenization workloads.

*Table 4.* **Tokenizers used in experiments for OpenAI's `tiktoken`.**

| Name | Code Name | Vocab Size | P99 Token Length |
|------|-----------|-----------|------------------|
| CL100K | `cl100k_base` | 100,256 | 16 |
| O200K | `o200k_base` | 199,998 | 19 |
| P50K | `p50k_base` | 50,280 | 14 |
| R50K | `r50k_base` | 50,256 | 14 |

We evaluate two input regimes, indicated by the **Eval** column in Table 5. In the `concat` setting, all documents in a dataset are concatenated into a single input stream, separated by newline characters, and tokenized as one continuous sequence. In the `per-doc` setting, each document is provided to the tokenizer as an independent input, and execution times are measured separately per document before aggregation. The results reported in the main text (Table 1) correspond to the `concat` setting, while Table 5 provides the full breakdown across both evaluation regimes.

*Table 5.* **Detailed throughput benchmarks.** **Base**: Original built-in BPE implementation. **Inc**: Our incremental BPE. **Eager**: "Inc" with eager output. **I/B**: Relative speedup of Inc/Base.

| Dataset | Eval | Tokenizer | Cache | Base (MiB/s) | Inc (MiB/s) | Eager (MiB/s) | I/B (×) |
|---------|------|-----------|-------|------|------|-------|-----|
| *Hugging Face's `tokenizers`* | | | | | | | |
| English | concat | CodeLlama | ✔ | 0.80 | 2.49 | 2.23 | **3.13** |
| English | concat | CodeLlama | ✘ | 0.80 | 2.45 | 2.19 | **3.05** |
| English | concat | DeepSeek-3.2 | ✔ | 1.52 | 1.54 | 1.51 | **1.01** |
| English | concat | DeepSeek-3.2 | ✘ | 1.42 | 1.48 | 1.44 | **1.04** |
| English | concat | GPT-OSS | ✔ | 1.89 | 1.90 | 1.88 | 1.00 |
| English | concat | GPT-OSS | ✘ | 1.89 | 1.92 | 1.89 | **1.01** |
| English | concat | Llama-3.1 | ✔ | 1.84 | 1.82 | 1.79 | 0.99 |
| English | concat | Llama-3.1 | ✘ | 1.87 | 1.87 | 1.85 | 1.00 |
| English | concat | Llama-4 | ✔ | 1.84 | 1.86 | 1.84 | **1.01** |
| English | concat | Llama-4 | ✘ | 1.86 | 1.88 | 1.85 | **1.01** |
| English | concat | Mistral-3 | ✔ | 1.83 | 1.83 | 1.80 | 1.00 |
| English | concat | Mistral-3 | ✘ | 1.84 | 1.83 | 1.80 | 0.99 |
| English | concat | Ouro | ✔ | 1.59 | 1.59 | 1.56 | 1.00 |
| English | concat | Ouro | ✘ | 1.51 | 1.57 | 1.50 | **1.04** |
| English | concat | Qwen-3 | ✔ | 1.52 | 1.59 | 1.54 | **1.05** |
| English | concat | Qwen-3 | ✘ | 1.34 | 1.50 | 1.45 | **1.12** |
| *OpenAI's `tiktoken`* | | | | | | | |
| English | concat | CL100K | — | 11.17 | 10.78 | 10.03 | 0.96 |
| English | concat | O200K | — | 6.21 | 6.12 | 5.98 | 0.99 |
| English | concat | P50K | — | 12.86 | 12.42 | 11.45 | 0.97 |
| English | concat | R50K | — | 12.83 | 12.38 | 11.49 | 0.96 |
| *Hugging Face's `tokenizers`* | | | | | | | |
| English | per-doc | CodeLlama | ✔ | 2.59 | 3.87 | 3.20 | **1.49** |
| English | per-doc | CodeLlama | ✘ | 2.59 | 3.89 | 3.20 | **1.50** |
| English | per-doc | DeepSeek-3.2 | ✔ | 2.04 | 2.15 | 2.09 | **1.05** |
| English | per-doc | DeepSeek-3.2 | ✘ | 1.83 | 2.04 | 1.91 | **1.11** |
| English | per-doc | GPT-OSS | ✔ | 2.78 | 2.80 | 2.73 | **1.01** |
| English | per-doc | GPT-OSS | ✘ | 2.77 | 2.81 | 2.75 | **1.01** |
| English | per-doc | Llama-3.1 | ✔ | 2.74 | 2.78 | 2.71 | **1.01** |
| English | per-doc | Llama-3.1 | ✘ | 2.78 | 2.81 | 2.72 | **1.01** |
| English | per-doc | Llama-4 | ✔ | 2.67 | 2.74 | 2.66 | **1.03** |

*Continued on next page...*

*Table 5.* **Detailed throughput benchmarks for Hugging Face's `tokenizers`.**

| Dataset | Eval | Tokenizer | Cache | Base (MiB/s) | Inc (MiB/s) | Eager (MiB/s) | I/B (×) |
|---|---|---|---|---|---|---|---|
| English | per-doc | Llama-4 | ✗ | 2.66 | 2.77 | 2.74 | **1.04** |
| English | per-doc | Mistral-3 | ✔ | 2.67 | 2.73 | 2.65 | **1.02** |
| English | per-doc | Mistral-3 | ✗ | 2.71 | 2.79 | 2.71 | **1.03** |
| English | per-doc | Ouro | ✔ | 2.26 | 2.29 | 2.21 | **1.01** |
| English | per-doc | Ouro | ✗ | 2.05 | 2.18 | 2.06 | **1.07** |
| English | per-doc | Qwen-3 | ✔ | 1.99 | 2.20 | 2.12 | **1.10** |
| English | per-doc | Qwen-3 | ✗ | 1.67 | 2.08 | 1.95 | **1.25** |
| *OpenAI's `tiktoken`* | | | | | | | |
| English | per-doc | CL100K | — | 11.54 | 11.19 | 10.35 | 0.97 |
| English | per-doc | O200K | — | 6.30 | 6.23 | 5.99 | 0.99 |
| English | per-doc | P50K | — | 13.43 | 12.98 | 11.98 | 0.97 |
| English | per-doc | R50K | — | 13.43 | 12.99 | 11.90 | 0.97 |
| *Hugging Face's `tokenizers`* | | | | | | | |
| Chinese | concat | CodeLlama | ✔ | 2.33 | 2.55 | 2.47 | **1.10** |
| Chinese | concat | CodeLlama | ✗ | 2.33 | 2.54 | 2.46 | **1.09** |
| Chinese | concat | DeepSeek-3.2 | ✔ | 1.52 | 1.41 | 1.32 | 0.93 |
| Chinese | concat | DeepSeek-3.2 | ✗ | 1.55 | 1.41 | 1.33 | 0.91 |
| Chinese | concat | GPT-OSS | ✔ | 1.71 | 1.84 | 1.73 | **1.08** |
| Chinese | concat | GPT-OSS | ✗ | 1.73 | 1.85 | 1.74 | **1.07** |
| Chinese | concat | Llama-3.1 | ✔ | 1.79 | 1.86 | 1.75 | **1.03** |
| Chinese | concat | Llama-3.1 | ✗ | 1.84 | 1.90 | 1.79 | **1.03** |
| Chinese | concat | Llama-4 | ✔ | 1.73 | 1.75 | 1.63 | **1.01** |
| Chinese | concat | Llama-4 | ✗ | 1.75 | 1.79 | 1.67 | **1.02** |
| Chinese | concat | Mistral-3 | ✔ | 1.73 | 1.80 | 1.71 | **1.04** |
| Chinese | concat | Mistral-3 | ✗ | 1.75 | 1.82 | 1.72 | **1.04** |
| Chinese | concat | Ouro | ✔ | 1.48 | 1.51 | 1.48 | **1.02** |
| Chinese | concat | Ouro | ✗ | 1.51 | 1.53 | 1.49 | **1.01** |
| Chinese | concat | Qwen-3 | ✔ | 1.59 | 1.65 | 1.54 | **1.04** |
| Chinese | concat | Qwen-3 | ✗ | 1.58 | 1.65 | 1.55 | **1.04** |
| *OpenAI's `tiktoken`* | | | | | | | |
| Chinese | concat | CL100K | — | 9.36 | 14.92 | 11.71 | **1.59** |
| Chinese | concat | O200K | — | 7.10 | 10.39 | 8.63 | **1.46** |
| Chinese | concat | P50K | — | 10.03 | 13.57 | 10.98 | **1.35** |
| Chinese | concat | R50K | — | 10.01 | 13.55 | 10.95 | **1.35** |
| *Hugging Face's `tokenizers`* | | | | | | | |
| Chinese | per-doc | CodeLlama | ✔ | 4.30 | 4.69 | 4.37 | **1.09** |
| Chinese | per-doc | CodeLlama | ✗ | 4.33 | 4.71 | 4.45 | **1.09** |
| Chinese | per-doc | DeepSeek-3.2 | ✔ | 1.94 | 1.76 | 1.60 | 0.91 |
| Chinese | per-doc | DeepSeek-3.2 | ✗ | 1.96 | 1.77 | 1.62 | 0.91 |
| Chinese | per-doc | GPT-OSS | ✔ | 2.25 | 2.41 | 2.22 | **1.07** |
| Chinese | per-doc | GPT-OSS | ✗ | 2.28 | 2.46 | 2.25 | **1.08** |
| Chinese | per-doc | Llama-3.1 | ✔ | 2.47 | 2.54 | 2.31 | **1.03** |
| Chinese | per-doc | Llama-3.1 | ✗ | 2.51 | 2.58 | 2.34 | **1.03** |
| Chinese | per-doc | Llama-4 | ✔ | 2.20 | 2.28 | 2.06 | **1.04** |
| Chinese | per-doc | Llama-4 | ✗ | 2.26 | 2.30 | 2.09 | **1.02** |
| Chinese | per-doc | Mistral-3 | ✔ | 2.31 | 2.44 | 2.22 | **1.06** |
| Chinese | per-doc | Mistral-3 | ✗ | 2.34 | 2.48 | 2.27 | **1.06** |
| Chinese | per-doc | Ouro | ✔ | 2.19 | 2.25 | 2.17 | **1.03** |

*Continued on next page...*

*Table 5.* **Detailed throughput benchmarks for Hugging Face's `tokenizers`.**

| Dataset | Eval | Tokenizer | Cache | Base (MiB/s) | Inc (MiB/s) | Eager (MiB/s) | I/B (×) |
|---|---|---|---|---|---|---|---|
| Chinese | per-doc | Ouro | ✘ | 2.23 | 2.30 | 2.20 | **1.03** |
| Chinese | per-doc | Qwen-3 | ✔ | 2.00 | 2.15 | 1.93 | **1.08** |
| Chinese | per-doc | Qwen-3 | ✘ | 1.97 | 2.15 | 1.94 | **1.09** |
| *OpenAI's* `tiktoken` | | | | | | | |
| Chinese | per-doc | CL100K | — | 9.75 | 16.00 | 12.37 | **1.64** |
| Chinese | per-doc | O200K | — | 7.37 | 11.09 | 9.09 | **1.50** |
| Chinese | per-doc | P50K | — | 10.56 | 14.50 | 11.52 | **1.37** |
| Chinese | per-doc | R50K | — | 10.55 | 14.45 | 11.46 | **1.37** |
| *Hugging Face's* `tokenizers` | | | | | | | |
| Code | concat | CodeLlama | ✔ | 0.82 | 2.37 | 2.21 | **2.88** |
| Code | concat | CodeLlama | ✘ | 0.81 | 2.34 | 2.17 | **2.87** |
| Code | concat | DeepSeek-3.2 | ✔ | 1.52 | 1.56 | 1.54 | **1.03** |
| Code | concat | DeepSeek-3.2 | ✘ | 1.45 | 1.54 | 1.48 | **1.07** |
| Code | concat | GPT-OSS | ✔ | 1.86 | 1.88 | 1.87 | **1.01** |
| Code | concat | GPT-OSS | ✘ | 1.88 | 1.88 | 1.87 | 1.00 |
| Code | concat | Llama-3.1 | ✔ | 1.78 | 1.80 | 1.78 | **1.02** |
| Code | concat | Llama-3.1 | ✘ | 1.82 | 1.86 | 1.82 | **1.02** |
| Code | concat | Llama-4 | ✔ | 1.84 | 1.85 | 1.83 | 1.00 |
| Code | concat | Llama-4 | ✘ | 1.86 | 1.85 | 1.83 | 0.99 |
| Code | concat | Mistral-3 | ✔ | 1.80 | 1.79 | 1.78 | 0.99 |
| Code | concat | Mistral-3 | ✘ | 1.79 | 1.80 | 1.79 | **1.01** |
| Code | concat | Ouro | ✔ | 1.53 | 1.57 | 1.54 | **1.02** |
| Code | concat | Ouro | ✘ | 1.48 | 1.55 | 1.50 | **1.05** |
| Code | concat | Qwen-3 | ✔ | 1.54 | 1.65 | 1.62 | **1.08** |
| Code | concat | Qwen-3 | ✘ | 1.38 | 1.59 | 1.54 | **1.15** |
| *OpenAI's* `tiktoken` | | | | | | | |
| Code | concat | CL100K | — | 8.16 | 8.50 | 7.96 | **1.04** |
| Code | concat | O200K | — | 5.49 | 5.51 | 5.41 | 1.00 |
| Code | concat | P50K | — | 8.29 | 8.90 | 8.05 | **1.07** |
| Code | concat | R50K | — | 8.43 | 8.85 | 8.01 | **1.05** |
| *Hugging Face's* `tokenizers` | | | | | | | |
| Code | per-doc | CodeLlama | ✔ | 2.11 | 3.62 | 3.17 | **1.72** |
| Code | per-doc | CodeLlama | ✘ | 2.11 | 3.62 | 3.16 | **1.72** |
| Code | per-doc | DeepSeek-3.2 | ✔ | 1.94 | 2.04 | 1.98 | **1.06** |
| Code | per-doc | DeepSeek-3.2 | ✘ | 1.81 | 2.00 | 1.91 | **1.11** |
| Code | per-doc | GPT-OSS | ✔ | 2.37 | 2.41 | 2.38 | **1.01** |
| Code | per-doc | GPT-OSS | ✘ | 2.45 | 2.51 | 2.47 | **1.02** |
| Code | per-doc | Llama-3.1 | ✔ | 2.36 | 2.46 | 2.38 | **1.04** |
| Code | per-doc | Llama-3.1 | ✘ | 2.43 | 2.51 | 2.43 | **1.03** |
| Code | per-doc | Llama-4 | ✔ | 2.36 | 2.37 | 2.37 | **1.01** |
| Code | per-doc | Llama-4 | ✘ | 2.30 | 2.37 | 2.33 | **1.03** |
| Code | per-doc | Mistral-3 | ✔ | 2.30 | 2.36 | 2.32 | **1.03** |
| Code | per-doc | Mistral-3 | ✘ | 2.34 | 2.41 | 2.36 | **1.03** |
| Code | per-doc | Ouro | ✔ | 2.01 | 2.08 | 2.02 | **1.03** |
| Code | per-doc | Ouro | ✘ | 1.87 | 2.01 | 1.93 | **1.07** |
| Code | per-doc | Qwen-3 | ✔ | 1.84 | 2.08 | 2.02 | **1.13** |
| Code | per-doc | Qwen-3 | ✘ | 1.64 | 2.03 | 1.92 | **1.24** |
| *OpenAI's* `tiktoken` | | | | | | | |

*Continued on next page...*

*Table 5.* **Detailed throughput benchmarks for Hugging Face's `tokenizers`.**

| Dataset | Eval | Tokenizer | Cache | Base (MiB/s) | Inc (MiB/s) | Eager (MiB/s) | I/B (×) |
|---|---|---|---|---|---|---|---|
| Code | per-doc | CL100K | — | 8.35 | 8.76 | 8.19 | **1.05** |
| Code | per-doc | O200K | — | 5.58 | 5.64 | 5.43 | **1.01** |
| Code | per-doc | P50K | — | 8.54 | 9.25 | 8.31 | **1.08** |
| Code | per-doc | R50K | — | 8.66 | 9.18 | 8.21 | **1.06** |

# I. Profiling Analysis of Tokenization Pipelines

To better understand the system-level bottlenecks in existing tokenization pipelines, we profile the execution using Linux `perf` and visualize the results with `inferno` (Gjengset, 2026). All flame graphs are collected from end-to-end tokenization runs, including normalization, pre-tokenization, and BPE, under the same configurations as the benchmark experiments in the main text.

For clarity of presentation, we apply light post-processing to the raw call stacks. Specifically, we retain only frames corresponding to the tokenization pipeline, truncate excessively deep stacks, and strip type and template information, preserving only function-level symbols and the structural hierarchy of call stacks. These transformations preserve the relative time distribution across pipeline stages while improving readability.

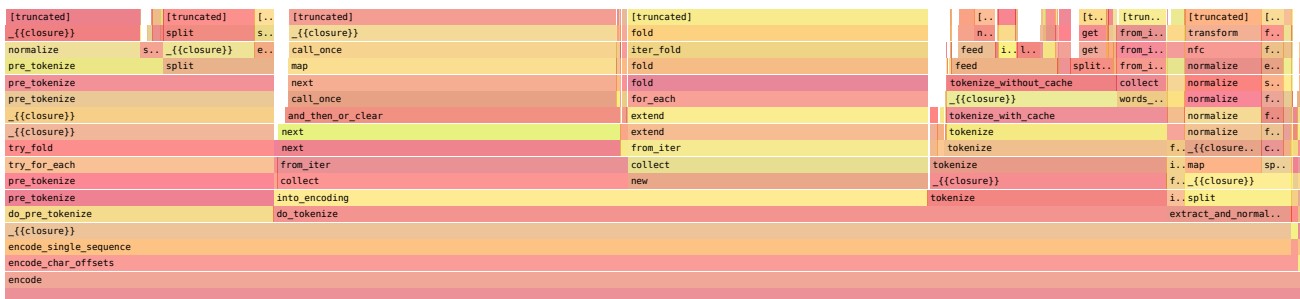

*Figure 5.* **Flame graph of `tokenizers` (Qwen-3) execution with our incremental non-eager implementation on the Code dataset.** The BPE merge phase (`tokenize_without_cache`) accounts for only 13.11% of the total execution time. The remaining time is dominated by normalization, pre-tokenization, and result construction cost (e.g., cloning token strings).

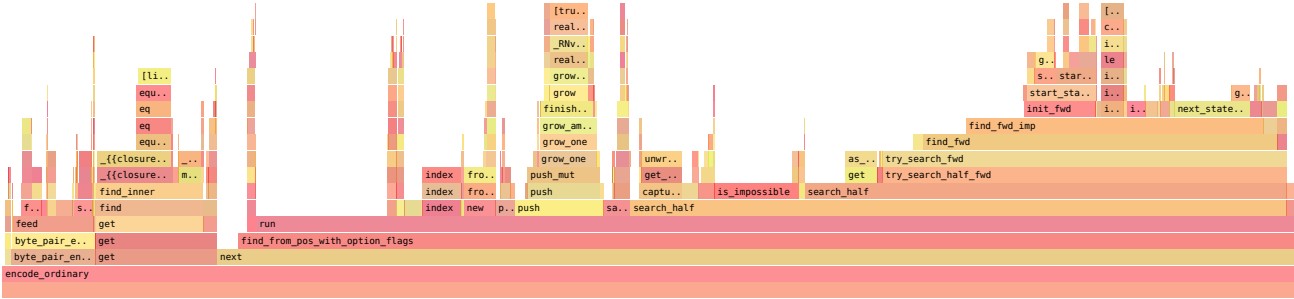

*Figure 6.* **Flame graph of `tiktoken` (O200K) execution with our incremental non-eager implementation on the Code dataset.** The regex matching phase (dominated by `find_from_pos_with_option_flags` and `run`) consumes approximately 80.25% of the total CPU time. In contrast, the actual BPE merge phase (`byte_pair_encode` on the left) accounts for only 6.45%. This indicates that, under this configuration, the pre-tokenization guardrail imposes a computational overhead exceeding 12× the cost of the core BPE merge stage.

Figure 5 presents a representative flame graph for Hugging Face's `tokenizers` library with the tokenizer Qwen-3, using our incremental non-eager BPE implementation on the Code dataset. The BPE merge phase itself (`tokenize_without_cache`) constitutes a minor fraction of the total runtime, highlighting that, in this pipeline, the overall throughput is primarily constrained by non-BPE components.

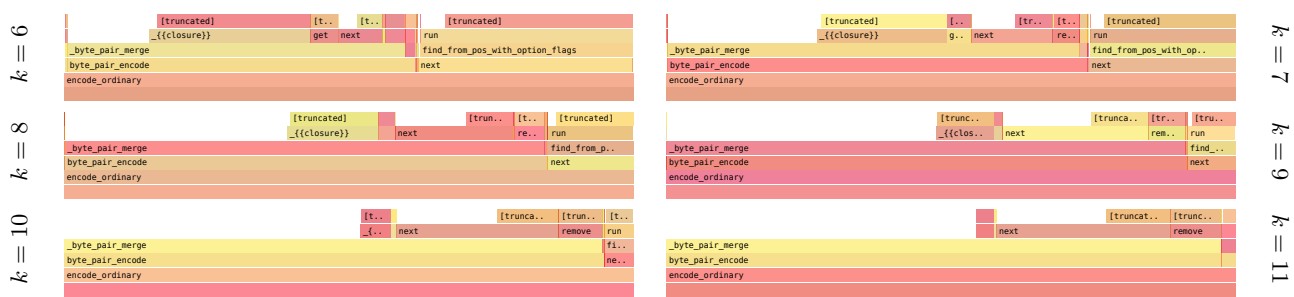

*Figure 7.* **Flame graphs of `tiktoken` using the original baseline implementation under pathological inputs with lengths from** $2^6$ **to** $2^{11}$**.** The BPE merge phase (`byte_pair_encode` on the left) becomes increasingly dominant as the input length grows, consistent with its $\mathcal{O}(n^2)$ time complexity, while the relative contribution of regex matching diminishes.

Figure 6 profiles `tiktoken` (O200K) when replacing its BPE stage with our incremental non-eager implementation on the Code dataset. It highlights that the bottleneck shifts to regex-based pre-tokenization, even when the BPE merge phase itself is asymptotically efficient.

Figure 7 profiles the original `tiktoken` baseline under pathological inputs of increasing length. As the input size grows, the BPE merge phase increasingly dominates the execution time, providing a fine-grained explanation for the throughput trends reported in Figure 3. These pathological inputs are not intended to model realistic workloads, but to isolate and stress-test the asymptotic behavior of the BPE merge stage.

## J. Comparison with Other Incremental BPE Implementations

In this section, we provide a detailed theoretical and empirical comparison between our algorithm and other existing incremental BPE implementations in Rust, specifically those found in the crate `bpe` of the GitHub repository `rust-gems` (GitHub, 2026).

For the discussion about time and space complexity, let $\Sigma$ be the size of the alphabet, $n$ be the input length, $m$ the vocabulary size, $t_i$ the length of the $i$-th token, and $t$ the maximum token length.

Within `rust-gems`, there are three main implementations of BPE as the methods of the structure `BytePairEncoding`. The `encode_via_bitfield` method utilizes a binary heap and does not support incremental processing, placing it outside the scope of discussion. The remaining two methods, `encode_via_table` and `encode_via_backtracking`, support incremental tokenization and are analyzed in the following.

All empirical tests and evaluations were performed by directly replacing the core BPE implementation in the `tiktoken` (OpenAI, 2026b) library. The tests were conducted on a Virtual Private Server (VPS), utilizing 8 cores of an Intel® Xeon® Gold 6448Y CPU situated on a single NUMA node, alongside a 16 GB slice of RAM.

### J.1. Theoretical Time Complexity

Both `encode_via_table` and `encode_via_backtracking` rely on a core underlying function: `is_valid_token_pair`. This function checks whether two tokens can be placed adjacently, ensuring that applying BPE to their concatenated string yields exactly the same two tokens.

Assuming the "last token" of a string is determined, if a new token can be validly placed adjacent to it, the "last token" of the newly formed string should exactly be this new token.

The validation process takes two tokens, iteratively un-merges either the first token to its successor or the second token to its predecessor according to the rule with the lower priority, and verifies if the token boundary survives. This verification takes $\mathcal{O}(t)$ time. Although it can be pre-computed or cached to achieve $\mathcal{O}(1)$ query time, the table will be $\mathcal{O}(m^2)$, which may not be preferable for production.

**Method 1: `encode_via_table`** This method processes the tokenization with an Aho–Corasick automaton. At each step, it iterates through the matching suffix tokens from longest to shortest for the current prefix, validating each suffix token

via `is_valid_token_pair`.

- **Complexity**: In the "best" case scenario, where the longest match is always the correct last token, it still requires at least one validation per step, yielding a lower bound of $\Omega(n)$ checks. In the worst case, all suffix tokens must be checked, resulting in $\mathcal{O}(nt^2)$ time complexity.

- **Eager Output**: Because this method maintains the prefix tree of tokens essentially, it can theoretically be adapted to support our proposed eager output mechanism.

**Method 2: `encode_via_backtracking`** This method employs a greedy strategy. It attempts to find the longest token among the prefixes of the un-tokenized string suffix, verifying it against the currently maintained token sequence via `is_valid_token_pair`. During the tokenization, it maintains a bit-field to record the positions that cause backtracking to avoid re-computation.

It is notable that the algorithm will backtrack if the current token sequence corresponds to a path from a leaf to the root in the prefix tree of tokens, which indicates no subsequent prefix token will form a valid sequence.

- **Complexity:** In the "best" case, where the greedy choice is always correct, the validation cost is amortized over the token length, yielding $\Omega(n)$. However, in the worst case, whenever the algorithm backtracks, it incurs the cost of validating all possible prefix tokens with the last token, yielding $\mathcal{O}(nt^2)$.

- **Eager Output:** As the algorithm is greedy, there is no trivial way to apply our eager output mechanism to this implementation.

Table 6 summarizes the complexity of these methods alongside our proposed algorithm. Our method incorporates a simple optimization where we first verify the longest suffix token before traversing the Suffix-Successor Tree, ensuring our "best" case also achieves an $\Omega(n)$ lower bound.

*Table 6.* **Overall complexity and feature comparison.**

| Method | Worst-Case | "Best"-Case | Eager Output |
|---|---|---|---|
| `encode_via_table` | $\mathcal{O}(nt^2)$ | $\Omega(n)^{\dagger}$ | Yes |
| `encode_via_backtracking` | $\mathcal{O}(nt^2)$ | $\Omega(n)$ | No |
| **Ours** | $\mathcal{O}(n\log^2 t)$ | $\Omega(n)$ | Yes |

$^{\dagger}$ Assume $\Omega(1)$ time for each check.

### J.2. Constructed Pathological Case

To approach the worst-case complexity, we designed an adversarial corner case. This construction deliberately traps algorithms relying on greedy or longest-prefix matching into massive validation and backtracking cycles.

First, to achieve a merge depth of $K = 4096$, we expand the base alphabet from single bytes to 2-byte pairs. To strictly prevent accidental cross-boundary merges, we construct base tokens $B_1, B_2, \ldots, B_K$ from $\mathcal{B}$:

$$\mathcal{B} = \{(i, j) \mid i, j \in \{0, 1, \ldots, 255\}, i < j\}.$$

The structural constraint of $\mathcal{B}$ guarantees that placing any two base tokens adjacent to each other will not trigger unintended cross-boundary merges.

Next, using these $K$ distinct base tokens, we construct a sequence $\mathcal{S}$ of $4K$ bytes:

$$\mathcal{S} = B_1 \cdots B_K B_K \cdots B_1.$$

We then strategically assign merge rule priorities to establish the trap: the absolute highest priority is assigned to the rule merging the two identical tokens located exactly at the sequence center, i.e., $(B_K, B_K)$. Subsequently, we inject deep nested merge rules originating from the center towards the two ends, i.e.:

$$(B_{K-1}, B_K), (B_{K-2}, B_{K-1}B_K), \ldots \quad \text{and} \quad (B_K, B_{K-1}), (B_K B_{K-1}, B_{K-2}), \ldots$$

establishing a merge depth of $\mathcal{O}(K)$.

The input string is constructed as a concatenation of 128 repetitions of $\mathcal{S}$. Under this pathological input, our method, with worst-case time complexity guarantees, required **37 ms** to tokenize the sequence. In stark contrast, `encode_via_table` and `encode_via_backtracking` required **15.3 s** and **23.9 s**, respectively.

## J.3. Space Complexity

**Initialization**  Our implementation utilizes a Square-Root Tiled Transition Table for $\mathcal{O}(1)$ transitions (detailed in Appendix F), requiring $\Theta(\sqrt{\Sigma})$ space per state. In contrast, `rust-gems` uses a double-array implementation of Aho-Corasick automata (Kanda et al., 2023), which performs transitions in $\mathcal{O}(1)$ time and normally requires $\mathcal{O}(1)$ space per state, but potentially up to $\mathcal{O}(\Sigma)$ in the worst case.

Additionally, our method requires pre-computing a Centroid Search Tree based on Centroid Decomposition for each Suffix-Successor Tree, leading to an initialization space complexity of $\mathcal{O}(\sum t_i) = \mathcal{O}(mt)$. `rust-gems` requires no more than $\mathcal{O}(m)$ auxiliary space.

**Tokenization and Eager Output**  During standard non-eager tokenization, both our method and the `rust-gems` implementations require $\mathcal{O}(n)$ space to record the token sequence.

However, space complexity behavior differs when applying our eager output mechanism. As detailed in Section 6, the algorithm maintains the state using a two-pointer approach. Token nodes may persist in the state even after they cease to belong to $\mathcal{P}$, because they lie in the window. Therefore, the space complexity relates to the maximum number of nodes simultaneously maintained by the two-pointer algorithm. While a theoretically safe upper bound is $\mathcal{O}(\min(n, mt))$, empirical evaluations show that the practical memory footprint is much smaller.

Table 7 demonstrates the maximum number of nodes in the window maintained by the two-pointer algorithm with the datasets described in Table 2.

*Table 7.* **Maximum size in the window during the two-pointer algorithm of eager output.**

| Tokenizer | English | Chinese | Code |
|-----------|---------|---------|------|
| P50K      | 26      | 35      | 96   |
| R50K      | 21      | 34      | 96   |
| CL100K    | 32      | 129     | 212  |
| O200K     | 31      | 129     | 208  |

## J.4. End-to-End Benchmarks

We evaluated the end-to-end tokenization speeds for `rust-gems` and ours. The tokenizer model P50K failed to run with `rust-gems`; thus, it was excluded from the benchmarks.

The median CPU times (in seconds) across standard datasets are presented in Table 8. The results are also demonstrated as a violin plot in Figure 8. The baseline implementation is the original BPE implementation in `tiktoken`. Our incremental approach consistently matches or outperforms the backtracking and table-based implementations.

*Table 8.* **End-to-end CPU time (seconds, median) compared with other incremental BPE implementations.** Lower is better.

| Model | Method | English | Chinese | Code |
|-------|--------|---------|---------|------|
| **R50K** | Baseline | 0.932 | 1.244 | 1.501 |
| | **Ours (inc)** | **0.925** | **0.897** | **1.369** |
| | Ours (eager) | 0.989 | 1.117 | 1.510 |
| | `via_backtracking` | 0.948 | 1.180 | 1.431 |
| | `via_table` | 1.004 | 1.243 | 1.566 |
| **CL100K** | Baseline | 1.068 | 1.374 | 1.535 |
| | **Ours (inc)** | **1.064** | **0.815** | **1.422** |
| | Ours (eager) | 1.135 | 1.050 | 1.515 |
| | `via_backtracking` | 1.090 | 1.048 | 1.441 |
| | `via_table` | 1.140 | 1.254 | 1.579 |
| **O200K** | Baseline | 1.768 | 1.729 | 2.082 |
| | **Ours (inc)** | **1.739** | **1.047** | **2.032** |
| | Ours (eager) | 1.818 | 1.308 | 2.081 |
| | `via_backtracking` | 1.782 | 1.275 | 2.087 |
| | `via_table` | 1.839 | 1.732 | 2.121 |

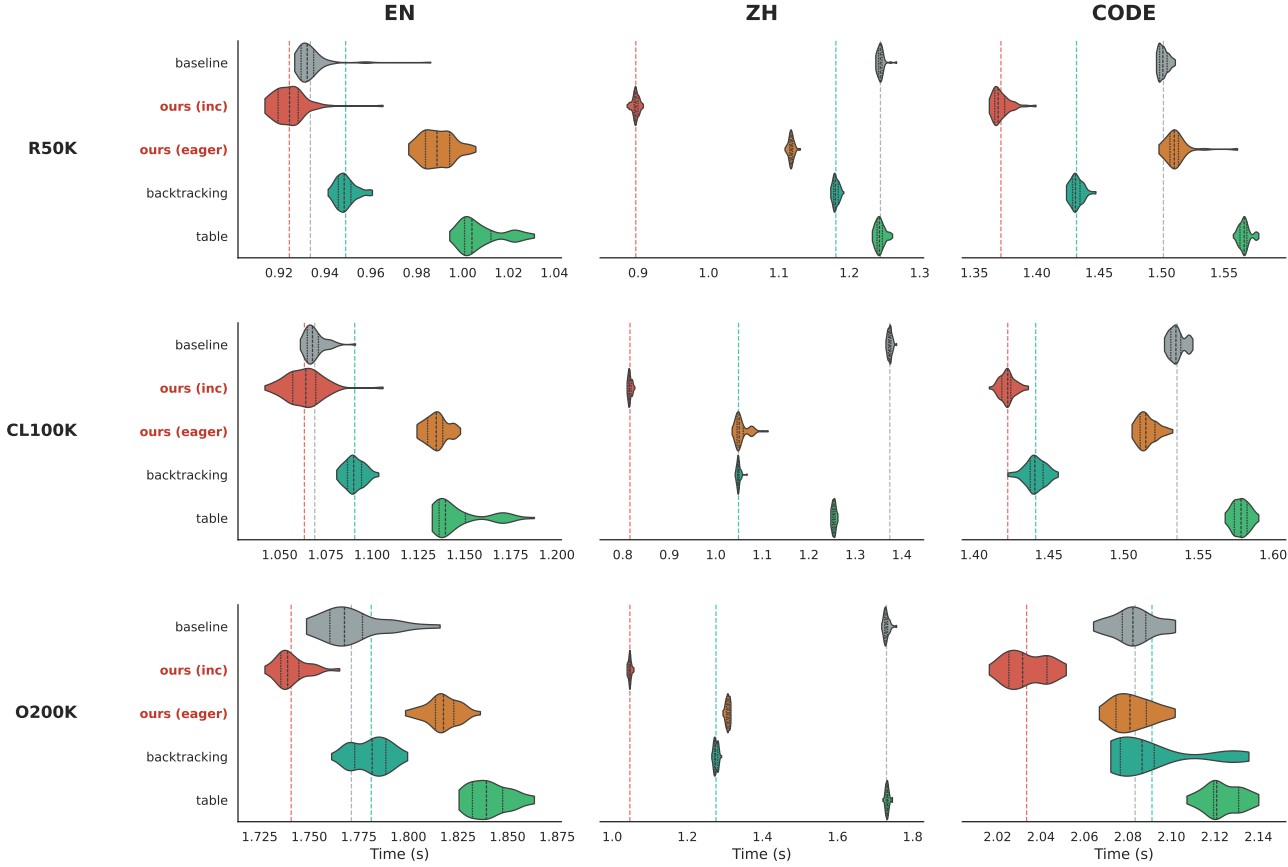

*Figure 8.* **Distributions of end-to-end execution times for different BPE implementations.** The violin plots illustrate the latency (in seconds, lower is better) across three tokenizers (R50K, CL100K, O200K) and three datasets (English, Chinese, Code). All methods were evaluated by replacing the core BPE in the `tiktoken` library. Specifically, baseline refers to the original BPE implementation of `tiktoken`, while `backtracking` and `table` correspond to the `encode_via_backtracking` and `encode_via_table` methods from `rust-gems`, respectively. Colored dashed vertical lines indicate the mean execution times for the baseline, **ours (inc)**, and `backtracking` implementations.

