# OpenReview forum: "Incremental BPE Tokenization"
_ICML.cc/2026/Conference — ICML 2026 spotlight_

### Official Review · Reviewer_89bL · 2026-03-12

**Soundness:** 4
**Presentation:** 4
**Significance:** 2
**Originality:** 3
**Overall Recommendation:** 5
**Confidence:** 4

**Summary:**

The paper gives an algorithm for BPE tokenization which takes worst-case time O(n log^2 t) time where n is the number of input bytes and t is the length of the longest token. The algorithm enables efficient tokenization of all prefixes of a string as well as incremental/streaming tokenization. An implementation is given which is significantly (~3x) faster than the BPE implementation of the HF Tokenizers library while being drop-in compatible.

**Compliance With Llm Reviewing Policy:**

Affirmed.

**Final Justification:**

See my rebuttal acknowledgement.

**Key Questions For Authors:**

1. What is the space complexity for the "eager" streaming algorithm?

Minor:

2. In line 139 (right), the notation for expression for the number of states is not clear.
3. In line 181 (right), the condition given is necessary but not sufficient. t itself must not be further merged after it is formed. Should this be mentioned? (It is mentioned later.)

**Limitations:**

1. The paper claims that "pre-segmentation can be algorithmically redundant" but this isn't the only benefit of pretokenization. Although it was introduced to bound the worst case time tokenization time under the naive quadratic algorithm, it has become useful as a way to control the tokenizer's behavior. (e.g. it is used to control the segmentation of numbers). In general, the pretokenization cannot just be dropped while maintaining the tokenizer's behavior. This becomes even more true when considering tiktoken's "ignore merges" path which skips BPE entirely when a pretoken matches a vocabulary type, allowing the formation of tokens which are not possible under BPE. Given this, it's not clear if the final product can still be reliable given the pretokenization may still exhibit pathological behavior.
2. The other issue is that it is not clear to me practically why faster tokenization is impactful (ignoring the removal of pathological behaviors which may persist due to other stages of the tokenization pipeline), given that one will generally invoke a model using tokens and the model is inevitably going to be much slower than the tokenizer.
3. The algorithms of  van Antwerpen & Neubeck (2024) and gweidart (2025) are not too complex, so it should be possible to show the proposed algorithm has better worst case time complexity. (My understanding was that both have time complexity no better than O(nt)), although I have not checked this carefully. Given these algorithms also come with fast implementations and have existed for a while, I believe it is important to show some advantage over them in some way, either with faster wall clock measurements (perhaps under some pathological tokenizer/input pair), or better time complexity.

**Strengths And Weaknesses:**

1. The presentation of the algorithm is very precise and logical. I was able to follow it clearly.
2. The algorithm seems pretty novel and uses some tricks to obtain better time complexity than existing algorithms. In particular, all of the existing fast algorithms for BPE run in O(nt) time or worse, so O(n log^2 t) is a potentially significant improvement, especially given the extremely long tokens found in some tokenizers.
3. The paper makes some sharp observations about the structure of BPE itself in order to achieve the stated complexity. These may be of interest beyond tokenization inference.
4. It is well-known that popular tokenization algorithms can "break" under pathological inputs, which this algorithm addresses, at least at the BPE level.

---

> ### Author Rebuttal · Authors · 2026-03-30
>
> We sincerely thank you for the thoughtful feedback and for recognizing the technical soundness and precision of our work. We appreciate the opportunity to address your questions regarding space complexity and the practical impact of our algorithm.
>
> ---
> ## `gweidart/rs-bpe` and `github/rust-gems`
> As mentioned in the reply to the review by Reviewer `8saT`, the core implementations in [`gweidart/rs-bpe`](https://github.com/gweidart/rs-bpe) are identical to those in [`github/rust-gems`](https://github.com/github/rust-gems). The following clarification will focus on the differences between our method and that of `github/rust-gems`.
>
> ---
> ## Space Complexity
> Let $\Sigma$ be the size of the alphabet, $n$ be the input length, $m$ the vocabulary size, $t_i$ the length of the $i$-th token, and $t$ the maximum token length.
> ### A. Initialization
>
> **1. Aho-Corasick Automata**
>
> Our implementation utilizes a "Square-Root Tiled Transition Table" for $O(1)$ transitions, and requires $\Theta(\sqrt{\Sigma})$ space per state, which is detailed in Appendix G.
>
> In contrast, `rust-gems` uses [`daac-tools/daachorse`](https://github.com/daac-tools/daachorse) ([arXiv 2207.13870](https://arxiv.org/abs/2207.13870)), a double-array implementation of Aho-Corasick automata. It also performs transitions in $O(1)$ time, normally requiring as little as $O(1)$ space per state, while potentially requiring up to $O(\Sigma)$ in the worst case per state.
>
> **2. Auxiliary Data**
>
> Our method requires pre-computing and storing structural information for each `SufSucTree` for centroid decomposition. To maximize query performance, we store this information per tree, leading to an initialization space complexity of $O(\sum{t_i}) = O(m t)$.
>
> In contrast, `rust-gems` requires no more than $O(m)$ additional auxiliary space related to the structural properties of tokens.
> ### B. Tokenization
> During tokenization, our method (without eager output) and implementations in `rust-gems` require recording the token sequence, whose space complexity is $O(n)$.
>
> However, the space complexity differs when applying eager output in our method or `encode_via_table` (a direct incremental implementation for all prefixes in `rust-gems`).
>
> **Q1: Eager Streaming Algorithm**
>
> As we are not limiting the vocabulary size and the maximum token length, the space complexity of eager output can reach beyond $\Omega(t)$.
>
> For example, if multiple long tokens share the same prefix and suffix, the "frontier" of pending tokens can grow continuously.
>
> While Berglund & van der Merwe (2023) show that incremental tokenization is viable with "finite lookahead", its theoretical upper bound of $O(m t)$ is impractically large.
>
> Thus, although the space overhead is theoretically independent of the input length $n$, a "practically safe" upper bound of space complexity of eager output is $O(\min(n, m t))$.
>
> Here is a table demonstrating **the maximum length of "frontier" tokens** with the real-world inputs with eager output:
> |Tokenizer|en|zh|code|
> |-|-|-|-|
> |P50K|26|35|96|
> |R50K|21|34|96|
> |CL100K|32|129|212|
> |O200K|31|129|208|
>
> ---
> ## L1: Pre-segmentation and Robustness
> We agree that pre-tokenization (e.g., regex-based or rule-based) serves functional roles.
>
> However, some functional effects of pre-tokenization can be achieved directly through vocabulary adjustments. For example, if the goal is to isolate digits, one can simply prohibit tokens mixing digits with other characters from appearing in the vocabulary.
>
> With our novel algorithm, we believe that a more streaming-friendly design of pre-tokenization will become preferable in the future.
>
> ---
> ## L2: Practical Impact of Faster/Incremental Tokenization
> While model inference indeed dominates standard batch processing time, the practical impact of our incremental tokenization primarily lies in **streaming scenarios** and minimizing **Time-To-First-Token (TTFT)**.
>
> In real-time applications (e.g., real-time voice assistants), standard tokenization acts as a blocking barrier, because it typically requires complete sequence chunks to process safely without pathological behaviors.
>
> By enabling incremental BPE and eager output, our algorithm eliminates this barrier. When fully integrated with streaming normalization, pre-tokenization and post-processing, it allows the tokenization to be fully pipelined with other stages, enabling continuous token delivery that overlaps with model execution to further optimize TTFT.
>
> ---
> ## L3: Comparison with `van Antwerpen & Neubeck (2024)` and `gweidart (2025)`
> Please refer to our response to **Reviewer 8saT**, where we provide a detailed
> comparison regarding these related works.
>
> ---
> ## Q2 & Q3: Minor
> - Line 139: This was a misuse of `\mathcal{\Theta}`. Thank you for pointing it
>   out. We will correct it in the revision.
> - Line 181: This was omitted primarily due to space constraints in the main
>   text. We will make adjustments in the revised version.

---

> > ### Author Rebuttal · Reviewer_89bL · 2026-04-03
> >
> > Thank you for your detailed response to my concerns. I consider this to be a significant contribution to the theory of BPE tokenization. Given the ubiquity of BPE tokenization in modern language modelling and the fact that all popular tokenizer libraries continue to use suboptimal algorithms. In practice, L does not tend to be large and BPE training generally will not create tokenizers with large L, so the pathological case will generally never occur in practice, which does limit the impact.
> >
> > Regarding pretokenization, I do want to mention that it is common to use the regex to also influence the digit grouping (e.g., left aligned groups of three digits). To my knowledge, it is not possible to express this purely using BPE merges.
> >
> > I have increased my score to 5.

---

> > > ### Author Response · Authors · 2026-04-04
> > >
> > > Thank you for updating the score and recognizing our theoretical contribution!
> > >
> > > We agree with your point on pre-tokenization. Certain rules like digit grouping are indeed hard to express purely through "proper" BPE merges, and `regex` currently fills the gap.
> > >
> > > However, our point is that these `regex`-based approaches act as a blocking barrier for streaming and may also become the runtime bottleneck in the tokenization pipeline.
> > >
> > > By providing a fast, streaming-capable baseline for the core BPE process, we hope to encourage the community to rethink tokenization pipeline design, inspiring a shift toward more streaming-friendly alternatives in the future.

---

### Official Review · Reviewer_NWXz · 2026-03-12

**Soundness:** 3
**Presentation:** 3
**Significance:** 3
**Originality:** 3
**Overall Recommendation:** 4
**Confidence:** 1

**Summary:**

This paper presents an incremental algorithm for Byte Pair Encoding (BPE) tokenization with provable worst-case complexity of O(n log² t). The core contribution is the introduction of two novel data structures — the Successor Forest and the Suffix-Successor Tree — which together enable efficient incremental updates to tokenization as new characters are appended to a string. The key theoretical result, the Monotonic Path Property (Theorem 4.2), characterizes the structural search space for the last token and enables constant-time verification of the Prefix Last-Token Condition via precomputed valid intervals. The algorithm is implemented using an Aho–Corasick automaton and Centroid Decomposition, and is evaluated across multiple tokenizer configurations including LLaMA-3.1, Gemma-3, and OpenAI tiktoken models, demonstrating significant throughput improvements over existing implementations such as HuggingFace tokenizers.

**Compliance With Llm Reviewing Policy:**

Affirmed.

**Final Justification:**

Weak accept

**Key Questions For Authors:**

This submission addresses incremental BPE tokenization from an algorithmic perspective, with contributions centered on novel data structures and provable complexity guarantees. While the work is technically sound, it is unclear whether this submission is the best fit for the LLM track at ICML, as the core contributions are more aligned with algorithms, data structures, and NLP systems engineering. The authors appear to assess the concept of incremental tokenization rigorously, and the authors attempt to outline a fundamental issue with existing BPE implementations. However, the connection to deep learning research — beyond tokenization being a preprocessing step — remains tenuous. The work may find a more receptive audience at venues such as MLSys, ACL System Track, or algorithms-focused conferences.

**Strengths And Weaknesses:**

**Originality:**

This paper presents a novel incremental algorithm for BPE tokenization with a well-defined worst-case complexity of O(n log² t). The introduction of the Successor Forest and Suffix-Successor Tree structures represents a meaningful algorithmic contribution. However, incremental and streaming tokenization has been partially explored in prior work, and the authors should more explicitly position their contributions against existing approaches such as tiktoken and SentencePiece to better clarify the novelty.

---

**Presentation:**

The paper is generally well-structured, with clear theoretical foundations and supporting proofs. However, the heavy reliance on formal notation and inductive proofs may hinder accessibility for a broader LLM audience. The authors are encouraged to provide more intuitive explanations and concrete examples alongside the theoretical derivations. Some sections, particularly those discussing non-properizable scenarios and monotonic path properties, feel overly dense and would benefit from illustrative figures or simplified walkthroughs.

---

**Soundness:**

The theoretical claims appear rigorous and well-supported. The complexity guarantees are clearly derived, and the robustness against pathological inputs (eliminating O(n²) degradation) is a practically valuable property. The experimental benchmarks demonstrate nearly 3× speedup over HuggingFace tokenizers in certain settings. However, the evaluation is somewhat narrow — gains are most pronounced in the absence of regex-based pre-tokenization, which is uncommon in modern LLM pipelines. The authors should report results under more realistic end-to-end settings and include a broader range of models and tokenizer configurations.

---

**Significance:**

From an LLM systems perspective, tokenization is a relatively lightweight component compared to model inference, and the practical impact may be limited in most production scenarios where pre-tokenization dominates the bottleneck. That said, the streaming output mechanism and robustness guarantees offer genuine value for real-time applications. The work is more likely to have impact as an engineering contribution than a core research advance for the LLM track. The authors' claim regarding "Green AI" benefits would be strengthened by concrete energy consumption measurements.

---

> ### Author Rebuttal · Authors · 2026-03-30
>
> Thank you for recognizing the algorithmic rigor and theoretical soundness of our
> work. We would like to clarify several factual points.
>
> ---
>
> ## Regarding `Summary`
>
> The mention of `Gemma-3` in the summary misinterprets its role in our paper.
>
> It is explicitly presented as a "non-properizable" case (refer to Appendix A.6
> and I.3) to explore theoretical boundaries between the standard BPE and
> `SentencePiece`-style BPE, rather than as a subject for end-to-end benchmarking
> with identical tokenization outputs.
>
> ## Regarding `Originality`
>
> The core innovation of this work is the $O(\log^2 t)$ per-byte incremental BPE
> algorithm itself.
>
> While `tiktoken` and `SentencePiece` are mentioned, they are "users" of the BPE
> merge logic and do not address the fundamental algorithmic complexity of the BPE
> merge process considered in this work.
>
> ## Regarding `Presentation`
>
> We believe the formal notation is necessary for theoretical rigor.
>
> To ensure accessibility, we specifically dedicated `Section 4.1 Intuition` in
> the main body to provide an intuitive overview of the core formalization and
> bridge the gap between formalization and practical understanding.
>
> ## Regarding `Soundness`
>
> > However, the evaluation is somewhat narrow — gains are most pronounced in the
> > absence of regex-based pre-tokenization, which is uncommon in modern LLM
> > pipelines.
>
> This might stem from a misinterpretation of the caption of Table 1:
>
> > Table 1. End-to-end throughput speedup.
> >
> > ...
> >
> > CodeLlama shows the highest gains due to the absence of regex-based
> > pre-tokenization.
>
> The note in Table 1 regarding `CodeLlama`'s performance gain was an objective
> observation of its lack of regex-based pre-tokenization.
>
> In `Appendix I.3. Tokenizers`, there is a full description of the
> pre-tokenization for each evaluated tokenizer.
>
> Even in the caption of Figure 1, we clearly state:
>
> > Our work focuses on making the BPE stage incremental without altering the
> > correctness and the surrounding stages.
>
> All experiments are designed to have the exact same tokenization as the baseline
> implementations for all frameworks, i.e. end-to-end evaluation.
>
> ## Regarding `Significance`
>
> As addressed in our response to Reviewer `89bL`, our work is critical for
> streaming scenarios and minimizing Time-To-First-Token (TTFT) by enabling
> pipeline parallelism.
>
> Beyond average-case throughput, our $O(\log^2 t)$ complexity solves a critical
> robustness issue. As shown in Figure 3, `tiktoken` degrades to $O(n^2)$ under
> pathological inputs, which can be exploited to cause system hangs. Our work
> provides a rigorous performance guarantee against worst-case scenarios.
>
> Regarding "Green AI", we agree that concrete energy measurements could further
> strengthen the claim. However, the primary contribution of this work is an
> improvement in worst-case algorithmic complexity, which directly reduces
> computational cost. Power-related considerations are therefore discussed in the
> Impact Statement rather than evaluated empirically in this work, as our
> algorithmic improvement primarily targets computational efficiency, which is the
> prerequisite for energy reduction.
>
> ---
>
> ## Regarding `Key Questions For Authors`
>
> We have a full paragraph talking about `Contributions` in `Introduction`, where
> we explicitly delineate our contributions to LLM research:
>
> > **Contributions.** We make the following contributions:
> >
> > - An incremental BPE algorithm is established with a strict, pioneering
> >   worst-case $O(\log^2 t)$ per-byte complexity, providing an exact solution
> >   with rigorous performance guarantees.
> > - The proposed eager output mechanism enables efficient, real-time streaming
> >   and pipelining with model inference.
> > - Our efficient implementation in Rust serves as a drop-in replacement that
> >   delivers an up to $\sim$3$\times$ speedup over current state-of-the-art
> >   tools.
> > - With our method, several stages of traditional tokenization pipelines (e.g.,
> >   regex-based pre-tokenization) can be safely skipped, revealing new
> >   acceleration potential.
> > - Our method provides native support for full-prefix tasks (e.g., FIM), by
> >   leveraging prefix trees of tokens to eliminate the need for heuristic
> >   truncation or re-computation.
>
> Beyond the contributions, a deeper structural understanding of BPE may
> contribute to improved interpretability of LLMs. Furthermore, efficient
> tokenization is fundamental to the study of LLM efficiency and inference
> systems, which are core topics in the ICML LLM track.

---

> > ### Author Rebuttal · Reviewer_NWXz · 2026-04-03
> >
> > Thank you for the response. I will increase my score.

---

> > > ### Author Response · Authors · 2026-04-03
> > >
> > > Thank you for reading our response and updating the score.

---

### Official Review · Reviewer_8saT · 2026-03-14

**Soundness:** 4
**Presentation:** 3
**Significance:** 3
**Originality:** 4
**Overall Recommendation:** 5
**Confidence:** 4

**Summary:**

This work introduces an algorithm for incremental byte-pair-encoding of text in $O(\log^2 t)$ per character, where $t$ is the maximum BPE token length. Incremental encoding means the algorithm operates in an online fashion processing the text one character at a time. At each point in time, the algorithm keeps track of the encoding of all prefixes of the currently processed prefix of the text. Processing the next character involves computing the BPE of the new prefix (which turns out to reduce just computing the last token of the new prefix), and this is done in $O(\log^2 t)$ time.

The work implements its algorithm in Rust, and compares it to the BPE algorithms present in HuggingFace’s tokenizers and OpenAI’s tiktoken. The work’s rust implementation is much faster when encoding large pieces of text, and avoids the pathological $O(n^2)$ behavior that tiktoken exhibits on certain inputs. However, in practice most production tokenizers use regex based pre-tokenization which breaks up text into smaller chunks that are tokenized, which means the paper’s algorithm when plugged into production tokenization pipelines only has better performance in certain scenarios, namely on Chinese text and code.

**Compliance With Llm Reviewing Policy:**

Affirmed.

**Final Justification:**

In light of the author's rebuttal, I have revised my score to a 5.

**Key Questions For Authors:**

No questions beyond the weaknesses I mentioned above. I just have a couple of additional comments:

* C1: Figure 2 seems to have the edge directions reversed.
* C2: I personally would have highlighted Lemma E.3 in the main body of the paper. This makes it much more intuitive what the monotonic path *is* and why it has the properties it has.
* C3: I personally would have also pulled in more details of the DFS linearization into the main body of the paper, as this is somewhat core to achieving $O(\\log^2 t)$.

**Limitations:**

L1: The primary limitation in my mind is that the authors do not compare their implementation against [https://github.com/github/rust-gems/blob/main/crates/bpe](https://github.com/github/rust-gems/blob/main/crates/bpe) and [https://github.com/gweidart/rs-bpe](https://github.com/gweidart/rs-bpe), which are also incremental BPE algorithms written in rust. I believe at least one of these implementations (bpe in particular) has $O(t)$ asymptotics, and since $t$ is small in practice, a-priori I think it’s possible that a $O(t)$ implementation would be faster than a $O(\\log^2 t)$ implementation. It would be nice to see an empirical comparison.

**Strengths And Weaknesses:**

Strengths and weaknesses are ordered by importance.

### Strengths

* S1: Achieving $O(\\log^2 t)$ is nontrivial and the algorithm introduced for doing so is quite elegant. I particularly appreciate the following components of the algorithm: the concept of a successor forest, Theorem 4.2 (Monotic Path Property), the DFS linearization and interval check approach to checking the prefix-last-token condition, and the usage of centroid decomposition. (+soundness, \+originality)

* S2: It’s cool that you implemented your solution in Rust and show it can outperform common production BPE algorithms. In particular I appreciate that your algorithm does not have the pathological O(n^2) behavior of tiktoken, which I have personally run into before. (+presentation, \+significance)

* S3: The ability to do eager output is cool\! (+presentation, \+significance)

* S4: I appreciate that you highlight the fact that the bottleneck in production tokenization systems is often the pre-tokenization stage. (+presentation, \+significance)

### Weaknesses

* W1: I would have liked to see both an empirical comparison of your implementation against [https://github.com/github/rust-gems/blob/main/crates/bpe](https://github.com/github/rust-gems/blob/main/crates/bpe) and [https://github.com/gweidart/rs-bpe](https://github.com/gweidart/rs-bpe). (-significance)

* W2: I would have liked to see a discussion of the theoretical properties of [https://github.com/github/rust-gems/blob/main/crates/bpe](https://github.com/github/rust-gems/blob/main/crates/bpe) and [https://github.com/gweidart/rs-bpe](https://github.com/gweidart/rs-bpe). For example, I skimmed these implementations and I believe that bpe should have a worst case O(t) performance per token. It would be nice to mention this in the paper, and maybe have a table comparing the properties of your new algorithm to bpe and rs-bpe. Properties of interest to me are: computational complexity, memory complexity, empirical performance, and whether eager mode is supported. (-presentation)

If the authors address W1 and W2/L1, I would happily raise my score to a 5\.

---

> ### Author Rebuttal · Authors · 2026-03-30
>
> We sincerely thank you for the thoughtful feedback and for recognizing the theoretical and practical value of our work. We appreciate the opportunity to clarify the theoretical properties and empirical performance compared to others.
>
> ---
> ## W2: `gweidart/rs-bpe` and `github/rust-gems`
> Upon inspection, the core implementations in `gweidart/rs-bpe` are identical to `github/rust-gems`. As `gweidart`'s repository *does not* mention `rust-gems` and *lacks a full commit history*, we will properly cite the GitHub blog and the original `github/rust-gems` repository in the Related Work and Benchmarks sections, and eliminate `gweidart/rs-bpe`.
>
> Nevertheless, in the benchmarks below, we compared `gweidart`'s repository, as thread-local LRU caches are added for short strings.
> ### A. Space Complexity
> The detailed analysis of space complexity is presented in the response to the review by Reviewer `89bL`. Please refer to that response.
> ### B. Overall Comparison
> |Method|Worst-case|"Best"-case|Eager output|
> |-|-|-|-|
> |`encode_via_table`|$O(n t^2)$|$\Omega(n t)$|Yes|
> |`encode_via_backtracking`|$O(n t^2)$|$\Omega(n)$|No|
> |Ours|$O(n\log^2 t)$|$\Omega(n)$|Yes|
> ### C. Constructed Corner-Case
> We constructed a corner-case vocabulary where tokens' predecessor/successor ancestor chain length can be $O(t)$. By encoding $0$ to $t$ as a subset of {$ (i\text{u8}, j\text{u8}) : i < j $}, we expanded the alphabet to construct deeply nested tokens like `(0, 1), ((0, 1), 2), ...`.
>
> **Code**: https://snips.sh/f/EAZy8jTvTI
>
> Our method executes in **milliseconds** (`37 ms`), while `rust-gems`'s `encode_via_backtracking` and `encode_via_table` take **seconds** (`23.9 s` and `15.3 s`).
> ### D. Theoretical Analysis
> In `github/rust-gems`, `encode_via_bitfield` uses a binary heap and does not support incremental processing, which falls outside the scope of our comparison.
>
> The other two, `encode_via_table` and `encode_via_backtracking`, can support incremental tokenization, and both rely on a core function: `is_valid_token_pair`.
>
> **1. `is_valid_token_pair`**
>
> This function checks whether two tokens can be placed adjacently, meaning that applying BPE to the concatenated string yields exactly the same two tokens.
>
> Assuming the "last token" of a string is determined, if a new token can be validly placed adjacent to it, the "last token" of the newly formed string should be exactly this new token.
>
> The two tokens iteratively expand towards their predecessors and successors based on merge priorities, verifying if the token boundary survives during the tokenization.
>
> This verification takes $O(t)$ time.
>
> **2. `encode_via_table`**
>
> This method processes all prefixes using an Aho-Corasick automaton.
>
> In each step, it iterates through the matching "suffix tokens" from longest to shortest for the current prefix, and validates each "suffix token" via `is_valid_token_pair`.
>
> - **Complexity:**
>   - "Best" case: the longest match is always the correct "last token". But it still requires at least one validation per step, yielding $\Omega(n t)$.
>   - Worst case: all "suffix tokens" must be checked, yielding $O(n t^2)$.
> - **Eager Output:** Because this method maintains the "prefix tree of tokens", it can be adapted to support our eager output mechanism.
>
> **3. `encode_via_backtracking`**
>
> This method uses a greedy strategy.
>
> It attempts to find the longest prefix token in the un-tokenized suffix, verifying it with the currently maintained token sequence via `is_valid_token_pair`.
>
> A notable issue arises if the current token sequence corresponds to a path from a leaf to the root in the "prefix tree of tokens". This indicates no subsequent prefix token will form a valid sequence, causing the algorithm to clear the bit-field and "backtrack".
> - **Complexity:**
>   - "Best" case: where the greedy choice is always correct, the cost of `is_valid_token_pair` is amortized over the token length, yielding $\Omega(n)$.
>   - Worst case: whenever the algorithm backtracks, it incurs the cost of validating all possible prefix tokens with the "last token", yielding $O(n t^2)$.
> - **Eager Output:** There is no naive way to apply our eager output mechanism to this implementation.
> ## E. "Best"-Case of Our Approach
> Our method incorporates a simple optimization: we first verify the longest "suffix token" before diving into the `SufSucTree`. Thus, our "best" case also achieves an $\Omega(n)$ lower bound complexity.
>
> ---
> ## W1 & L1: End-to-End Benchmarks
> - Full Results: https://snips.sh/f/iQxd3xPzXt
>   - CPU Time in seconds. Median.
>   - Tested by replacing core BPE in `tiktoken`.
>   - `P50K` fails to run with `rust-gems`.
> - Violin Plots: https://imgur.com/a/q7GBwJd
> ### Results (`CL100K`, Partial)
> |Method|en|zh|code|
> |-|-|-|-|
> |Baseline|1.07|1.37|1.53|
> |Ours (inc)|1.06|0.81|1.42|
> |`via_backtracking`|1.09|1.05|1.44|
> |`gweidart`|1.21|1.83|1.58|
>
> ---
> ## Other Comments (C1, C2, C3)
> We will refine the paper based on the comments. The comparative analysis above will also be included in the Appendix.

---

> > ### Author Rebuttal · Reviewer_8saT · 2026-04-03
> >
> > Thank you for addressing my concerns. I've now raised my score to a 5!

---

> > > ### Author Response · Authors · 2026-04-03
> > >
> > > Thank you for updating the score. We truly appreciate your constructive review and your support for our work!

---

### Decision · Program_Chairs · 2026-04-30

**Decision:**

Accept (spotlight)

**Comment:**

This paper proposes an incremental BPE tokenization algorithm with worst-case $O(n \log^2 t)$ complexity, supported by novel data structures and accompanied by a practical implementation. Overall, the reviewers agree that the work is technically strong, original, and well-grounded, with particular strengths in its rigorous analysis and algorithmic design. Given the reviewer support, solid technical novelty, and demonstrated practical relevance in streaming and worst-case robustness settings, I recommend acceptance.